# The RNA binding protein DND1 is elevated in a subpopulation of pro-spermatogonia and targets chromatin modifiers and translational machinery during late gestation

Victor A. Ruthig[1,2]*, Talia Hatkevich[2], Josiah Hardy[2], Matthew B. Friedersdorf[3], Chloé Mayère[4,5], Serge Nef[4,5], Jack D. Keene[3], Blanche Capel[2]*

**1** Sexual Medicine Lab, Department of Urology, Weill Cornell Medicine, New York, New York, United States of America, **2** Department of Cell Biology, Duke University Medical Center, Durham, North Carolina, United States of America, **3** Department of Molecular Genetics and Microbiology, Duke University Medical Center, Durham, North Carolina, United States of America, **4** Department of Genetic Medicine and Development, University of Geneva, Geneva, Switzerland, **5** iGE3, Institute of Genetics and Genomics of Geneva, University of Geneva, Geneva, Switzerland

* vruthig@hawaii.edu (VAR); blanche.capel@duke.edu (BC)

## Abstract

DND1 is essential to maintain germ cell identity. Loss of *Dnd1* function results in germ cell differentiation to teratomas in some inbred strains of mice or to somatic fates in zebrafish. Using our knock-in mouse line in which a functional fusion protein between DND1 and GFP is expressed from the endogenous locus (*Dnd1^GFP*), we distinguished two male germ cell (MGC) populations during late gestation cell cycle arrest (G0), consistent with recent reports of heterogeneity among MGCs. Most MGCs express lower levels of DND1-GFP (DND1-GFP-lo), but some MGCs express elevated levels of DND1-GFP (DND1-GFP-hi). A RNA-seq time course confirmed high *Dnd1* transcript levels in DND1-GFP-hi cells along with 5-10-fold higher levels for multiple epigenetic regulators. Using antibodies against DND1-GFP for RNA immunoprecipitation (RIP)-sequencing, we identified multiple epigenetic and translational regulators that are binding targets of DND1 during G0 including DNA methyltransferases (Dnmts), histone deacetylases (Hdacs), Tudor domain proteins (Tdrds), actin dependent regulators (Smarcs), and a group of ribosomal and Golgi proteins. These data suggest that in DND1-GFP-hi cells, DND1 hosts coordinating mRNA regulons that consist of functionally related and localized groups of epigenetic enzymes and translational components.

## Author summary

DND1, a RNA-binding protein, has emerged as a regulator of germ cell identity in vertebrates. In the absence of *Dnd1*, germ cells are lost or tend to differentiate into teratomas or somatic cell types in mice and zebrafish, pointing to an essential role of this post-transcriptional regulator in maintaining the unique identity of germ cells. Using a mouse line

**Data Availability Statement:** The raw hiloRNA-seq and RIP-seq data is available on GEO (GSE197610) as a super-series. The complete script for all data analysis has also been deposited in GEO as a singular simple text file in Rmarkdown syntax along with all the supporting, intermediate, and Rstudio data files used in the analysis. These scripts are complete but contain redactions to protect the cybersecurity of the Duke Computing Cluster and the Scientific Computing Unit at Weill Cornell Medicine, and therefore will need adjustments to function on other computing clusters and with Rstudio sessions. The unprocessed matrix counts and processed Loupe File from the aggregated E16.5 scRNA-seq re-analysis have also been deposited on GEO (GSE197610) as part of the same super-series and are linked to the raw scRNA-seq data published in Law et al. 2020 which is also on GEO (GSE124904) https://doi.org/10.1038/s41467-019-10596-0. The Dnd1Ter-seq dataset published in Ruthig et al. 2019 was deposited with that publication on GEO (GSE132719) https://doi.org/10.1242/dev.175950.

**Funding:** This work was supported by: the National Institutes of Health National Institute of General Medical Sciences (NIH-NIGMS) [F32GM129956] as salary support to VAR; a Duke Josiah Charles Trent Memorial Foundation grant to VAR; a Duke Cancer Institute (DCI) Pilot Grant to BC; the National Institutes of Health Eunice Kennedy Shriver National Institute of Child Health and Human Development (NIH-NICHD) [R37HD039963] to BC; and funding from the Duke Summer Research Opportunity Program (SROP) to JH. The funders had no role in study design, data collection and analysis, decision to publish, or preparation of the manuscript.

**Competing interests:** The authors have no competing interests to disclose.

carrying a GFP-tag on endogenous *Dnd1* (DND1$^{GFP}$), we discovered distinct populations of male germ cells expressing high and low levels of DND1 during late gestation. RNA sequencing identified differences between these populations. DND-GFP-hi cells initially express elevated levels of transcripts encoding pluripotency genes which they later down-regulate. Transcripts encoding translational machinery and epigenetic regulators are higher in DND1-GFP-hi cells throughout gestation and frequently map as RNA-binding targets from DND1 RNA regulons as discovered by RNA immunoprecipitation (RIP)-sequencing. MGCs expressing high levels of DND1 are much less likely to express a phagocytic signal on their cell surface and are more likely to express transcripts for genes associated with the transition to pro-spermatogonial fate.

## Introduction

Dead end 1 (*Dnd1)* is a RNA binding protein (RBP) that is essential for the maintenance of the germ cell population in multiple vertebrate species [1, 2]. For example, in many inbred mouse strains, most germ cells carrying the *Dnd1$^{Ter/Ter}$* mutation undergo apoptosis soon after speci-fication [3, 4]. On some genetic backgrounds, *Dnd1$^{Ter/Ter}$* male germ cells (MGCs) that escape apoptosis often differentiate into teratomas with many embryonic cell types represented [3, 5]. This is broadly consistent with findings in zebrafish, where some *dnd1* mutant germ cells die, while others undergo differentiation to somatic lineages during migration to the gonad [6]. These studies suggest that DND1 plays a critical role in maintaining germ cell identity by pro-moting survival and suppressing differentiation to somatic fate.

Suppression of somatic fate occurs through epigenetic mechanisms in C. elegans [7, 8] and is believed to occur through similar mechanisms in mammals. During migration in mammals, germ cell DNA is depleted of methylation [9, 10] through a combination of active oxidation via the Tet enzymes and passive loss of 5mC through rapid proliferation [11]. At embryonic day (E)8.5, the stable repressive mark H3K9me2 is eliminated and H3K27me3 is acquired at bivalent developmental loci [12, 13]. The capacity to generate embryonic germ cell lines is high at E12.5, suggesting that many embryonic differentiation pathways are "open" at this stage.

During the last third of fetal life until a few days after birth (E14.5 to about postnatal day (P) 2), MGCs enter a long period of cell cycle arrest in G0 [14–16]. During this period MGCs undergo extensive epigenetic reprogramming that correlates with a reduction in their potential to give rise to teratomas [5] or to embryonic germ cell lines [17]. By E16.5, when *Dnmt3a* and *Dnmt3l* are transcribed, re-methylation begins [18–20]. By birth, DNA is hypermethylated [19, 20]. ATAC-seq and ChIP-seq for 6 histone marks documented chromatin accessibility changes between E13.5-P1, mainly associated with the acquisition of repressive histone marks includ-ing H3K27me3 and H3K9me2/3 [21] often around transposable elements [20]. It is not known exactly when these changes occur or whether they are uniform across the population, but sev-eral lines of evidence suggest that DND1 is involved.

First, a prior study showed that DND1 promotes cell cycle arrest in G0 by mediating tran-script stabilization of the methyltransferase *Ezh2* which deposits the repressive mark H3K27me3 at the genetic locus of the activating cell cyclin *Ccnd1* [22]. In our previous study of the *Dnd1$^{Ter/Ter}$* mutants, we found that transcripts for multiple epigenetic modifiers were down-regulated relative to levels in wild type MGCs [23]. However, it was not clear whether this was a direct or indirect effect of DND1 loss. By conducting digestion optimized RNA immunoprecipitation (DO-RIP)-seq against DND1 in HEK293 cells, we identified the DND1

binding motif, DAUBAW ([A/G/U]AU [C/G/U]A[A/U]), and many in vitro DND1 targets [23]. Our binding motif and in vitro target data complimented a similar study on DND1 [2]. Although some targets in HEK293 cells were chromatin modifiers, many pluripotency and epigenetic regulators significantly altered in $Dnd1^{Ter/Ter}$ mutants are not expressed in HEK293 cells [23].

Functionally, DND1 has been shown to stabilize some target transcripts by blocking micro-RNA-mediated destruction [24], and destabilize other target transcripts by recruitment to processing (P) bodies as part of the CNOT (carbon catabolite repressor 4-negative on TATA) complex [1, 2]. Target transcript degradation via the CNOT complex relies on protein-protein interactions between DND1 and a second RNA binding protein NANOS2, which can modify DND1 target selection [1, 25]. Recent structural analysis has shown that motif recognition prompts a two-step RNA binding mechanism resulting in oppositional binding by the two tandem RNA recognition motifs (RRM) of DND1 in a clamp fashion, while the double-stranded recognition motif (DSRM) remains exposed, putatively to be bound by a protein partner such as NANOS2 [26].

Absence of an available antibody against DND1 interfered with analysis of its role in MGC fate commitment in vivo. To solve this problem, we used CRISPR/Cas9 editing to establish a knock-in mouse line expressing a $Dnd1^{GFP}$ fusion allele from the endogenous locus. In both male and female fetal development, levels of DND1$^{GFP}$ expression varied among germ cells [27], consistent with heterogeneity in the germ cell population reported by multiple groups [28–30]. At E14.5, E16.5, and E18.5 we detected DND1$^{GFP}$-hi (DND1-GFP-hi) and DND1$^{GFP}$-lo (DND1-GFP-lo) expressing populations. We show that these populations have distinct transcriptomes with high levels of phosphatidylserine on the surface of DND1-GFP-lo cells, suggesting reduced viability. Using the GFP-tag on DND1 to perform RNA immunoprecipitation (RIP) assays at E14.5, E16.5, and E18.5, we show that DND1 binds multiple RNAs encoding cell cycle genes, epigenetic regulators and genes associated with the Golgi and vesicle transport, all of which are present at higher levels in DND1-GFP-hi cells. These data suggest that high levels of DND1 play a direct role in promoting viability and reprogramming of MGCs during cell cycle arrest by binding to and regulating the translation of transcripts that encode critical chromatin modifiers and translational machinery.

## Results

### During late gestation subpopulations of MGCs express high or low levels of Dnd1

We previously used CRISPR/Cas9 to target the endogenous allele of $Dnd1$ and insert $eGfp$ fused in frame to the 5' end of the $Dnd1$ gene ($Dnd1^{GFP}$). This allele revealed heterogeneity in male germ cells (MGCs) during stages preceding mitotic arrest (E11.5-E14.5) [27], consistent with results from other labs [28–30]. We used DND1-GFP to isolate MGCs at later fetal stages. Using fluorescence activated cell sorting (FACS) we isolated DND1-GFP-positive cells at E14.5, E16.5, E18.5, and P3. To our surprise FACS revealed distinct DND1-GFP-lo and DND1-GFP-hi populations at E14.5, E16.5 and E18.5 (Fig 1A and 1B). At E14.5, the DND1-GFP-hi population accounts for <1% of MGCs. Their numbers peak at E16.5, when DND1-GFP-hi cells represent 12% of total MGCs, and decline again at E18.5, when the DND1-GFP-hi population accounts for only 5% of MGCs (S1 Table). FACS analysis of MGCs from $Oct4$-GFP testes at E18.5 did not reveal high and low subpopulations (Fig 1B). As MGCs remain in G0 cell cycle arrest throughout these stages, differences in DND1 levels cannot be related to cell cycle differences.

To discount the possibility that high and low levels of DND1 were an artifact of cell dissociation and isolation via FACS, we investigated whether MGCs expressing high and low levels of

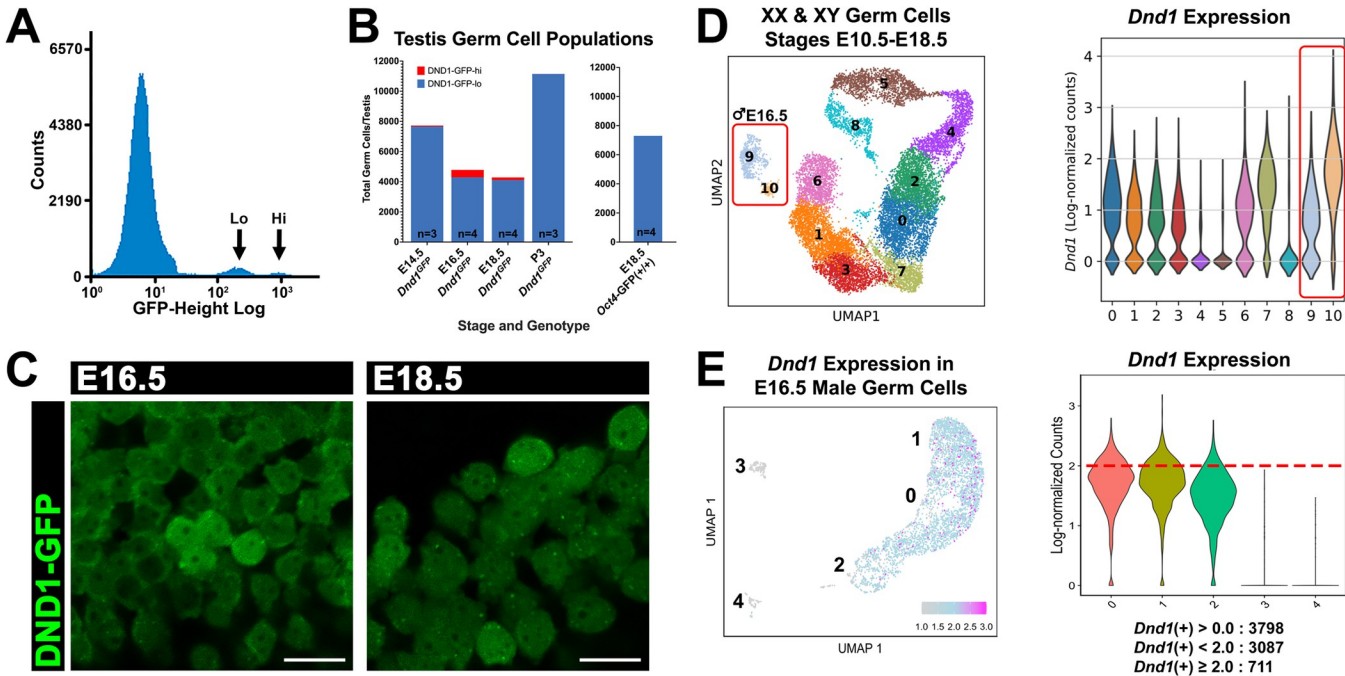

**Fig 1. Fetal male germ cells express low or high levels of DND1-GFP and *Dnd1*. A:** FACS histogram showing one DND1-GFP-negative, and two DND1-GFP-positive populations, representing DND1-GFP-lo and DND1-GFP-hi cells at E18.5. **B:** FACS quantification for the number of DND1-GFP-lo and DND1-GFP-hi germ cells from E14.5, E16.5, E18.5, P3 DND1-GFP testes (n = 3,4,4,3 respectively). FACS analysis of germ cells from E18.5 *Oct4*-GFP testes (n = 4) did not show GFP low and high subpopulations. **C:** Whole mount confocal imaging of unstained freshly dissected E16.5 and E18.5 DND1-GFP testes showing heterogenous endogenous DND1-GFP fluorescence. Scale bar = 20 μm. **D:** scRNA-seq UMAP for XX and XY germ cell populations from E9.0-E16.5 [32] and violin plot showing DND1 expression levels in clusters 0–10. Red box surrounds E16.5 male germ cell populations represented by clusters 9 and 10. Corresponding violin plots show that clusters 9 and 10 contain cells expressing low and high levels of *Dnd1* respectively. **E:** re-analysis of E16.5 male germ cell scRNA-seq from [31]. UMAP colored to show low to high levels of *Dnd1* expression (grey to blue to pink). Violin plot showing levels of *Dnd1* expression in clusters 0, 1, 2, 3, and 4, with a level of 2.0 denoted by red broken line. Quantification of total number of *Dnd1*-expressing cells with expression ≥2.0 = 711 (of 3798 total DND1+ cells).

DND1 were visible within live testis cords. We mounted E16.5 and E18.5 *Dnd1*$^{GFP}$ freshly dissected testes on a slide and used confocal microscopy to qualitatively explore levels of endogenous DND1-GFP. DND1-GFP-lo and DND1-GFP-hi cells were evident in all images captured from live samples (Figs 1C and S1) and fixed samples (S2A Fig). To quantitatively assess endogenous DND1-GFP levels, we quantified the endogenous DND1-GFP in DND1-GFP-hi (S2B Fig) and DND1-GFP-lo (S2C Fig) cells in fixed testes, normalizing to DNA (S2D Fig). DND1-GFP-lo cells appear as a continuum. However, in quantitative and FACS analysis, a distinct DND1-GFP-hi population emerged. In both live and fixed testis cord samples, we found that DND1-GFP-hi cells were sometimes found in clusters and were sometimes dispersed in cords. To rule out the possibility that high/low expression was an artifact of the knock-in, we investigated 2 single cell datasets from E16.5 gonads [31, 32]. Both datasets defined at least one subpopulation of DND1-GFP-hi cells based on native expression from the locus (Fig 1D and 1E). To further ensure that the DND1-GFP-hi population is not due to an over-expression of the modified *Dnd1*$^{GFP}$ allele, we isolated DND1-GFP-lo and DND1-GFP-hi populations from *Dnd1*$^{GFP/+}$ heterozygotes and performed RT-qPCR for expression from each allele. Similar to our previous analyses, we see a higher mean expression of *Dnd1* relative to *Gapdh* in the DND1-GFP-hi cells, as compared to the DND1-GFP-lo cells (S2E Fig, p-value ≤0.01). However, when comparing mean expression of *eGfp* relative to *Dnd1*, we found no significant difference in DND1-GFP-hi vs. DND1-GFP-lo cells (S2F Fig, p-value >0.05).

## Although DND1-GFP-lo and DND1-GFP-hi cells share 62–84% of their transcriptomes, DND1-GFP-hi cells express higher levels of Dnd1, pluripotency markers and components of the OxPhos pathway

To determine whether the transcriptome differed between cells expressing high and low levels of DND1, we isolated 4 independent replicates of paired DND1-GFP-lo and DND1-GFP-hi cells by FACS at each of two timepoints (E16.5 and E18.5, S1 Table) and performed RNA-seq. Transcriptome data within each set of 4 samples (E16.5 DND1-GFP-lo, E16.5 DND1-GFP-hi, E18.5 DND1-GFP-lo, or E18.5 DND1-GFP-hi) was highly correlated with all replicate samples from that set, scoring a Pearson correlation coefficient of $\geq 0.88$. Overall, among all high and low samples the lowest level of correlation was 0.76, indicating similarities among all these germ cell populations (S3 Fig). Global comparison of the hiloRNA-seq datasets at E16.5 revealed that out of the expression of 11,937 total transcripts, 7,271 (61%) transcripts are shared between DND1-GFP-lo and DND1-GFP-hi cells. At E18.5, the populations are more similar transcriptionally: out of 12,208 total transcripts, 10,198 (84%) are shared (Fig 2A). Shared genes include transcriptional regulators, metabolite interconversion enzymes, and protein modifiers and modulators.

However, despite the high level of similarity, there were significant differences in the transcriptomes of DND1-GFP-lo and DND1-GFP-hi cells at both E16.5 and E18.5. A large group of differentially expressed genes (DEGs) were detected (expression TPM $\geq 5.0$ in lo or hi cells, L2FC $\geq 1.0$ or $< -1.0$ for hi versus lo, p-value $\leq 0.05$) in DND1-GFP-lo or DND1-GFP-hi cells at E16.5 and E18.5, with a higher number of differentially expressed genes in DND1-GFP-hi cells (Fig 2B and S2 Table). At E16.5 DND1-GFP-lo cells have 1,032 over-expressed genes, while DND1-GFP-hi cells have 3,634 over-expressed genes. At E18.5, only 154 genes were significantly elevated in DND1-GFP-lo cells compared to 1,856 genes significantly elevated in DND1-GFP-hi cells (Fig 2B). Only 35 differentially expressed genes were shared between DND1-GFP-lo cells at E16.5 and E18.5. Whereas in DND1-GFP-hi cells, 1,709 differentially expressed genes were common between E16.5 and E18.5 (Fig 2B), showing a strong conservation of the differentially expressed genes in DND1-GFP-hi cells over time.

As a first analysis, we investigated whether *Dnd1* and other markers of germline stem cells are expressed at different levels in DND1-GFP-lo and DND1-GFP-hi cells. DND1-GFP-hi cells showed approximately 7-8-fold higher levels of *Dnd1* transcripts, confirming expectations, although levels of *Dnd1* expression decline between E16.5 and E18.5 (Fig 2C). At E16.5, DND1-GFP-hi cells still had higher levels of transcripts of other markers associated with the germline stem cell population, including *Nanos2*, *Dazl*, *Ddx4*, *Dppa3*, *Tfap2c*, and *Sox2*, all of which sharply declined between E16.5 and E18.5. Of these genes, only levels of *Dnd1*, *Dazl* and *Ddx4* remain elevated at E18.5 (Fig 2C). Gene ontology (GO) analysis on differentially expressed genes (DEGs) identified differences in energy metabolism between the DND1-GFP-lo and DND1-GFP-hi populations. Most striking of these metabolic differences was the increase of oxidative phosphorylation in DND1-GFP-hi cells (Fig 2D). DND1-GFP-lo cells expressed higher levels of transcripts of genes associated with signaling, such as *Bmp5* and *Wnt4* (Fig 2E). In support of this DEG data, re-analysis of the E16.5 MGC scRNA-seq data [31] showed that many of the same genes related to epigenetic and pluripotency regulation (*Brdt*, *Dnmt3a*, *Mael*, *Piwil4*) had elevated expression in the same clusters as DND1-hi cells (clusters 0,1; S4 Fig), whereas other DEGs that were more highly expressed in our DND1-GFP-lo cells (*Kdm6b*, *Lefty2*) had elevated expression in the same cluster as DND1-lo cells (cluster 2; S4 Fig). Some epigenetic and pluripotency regulation genes highly expressed in DND1-GFP-hi cells (*Dazl*, *Dppa5a*) did not follow this pattern in the scRNA-seq data (S4 Fig).

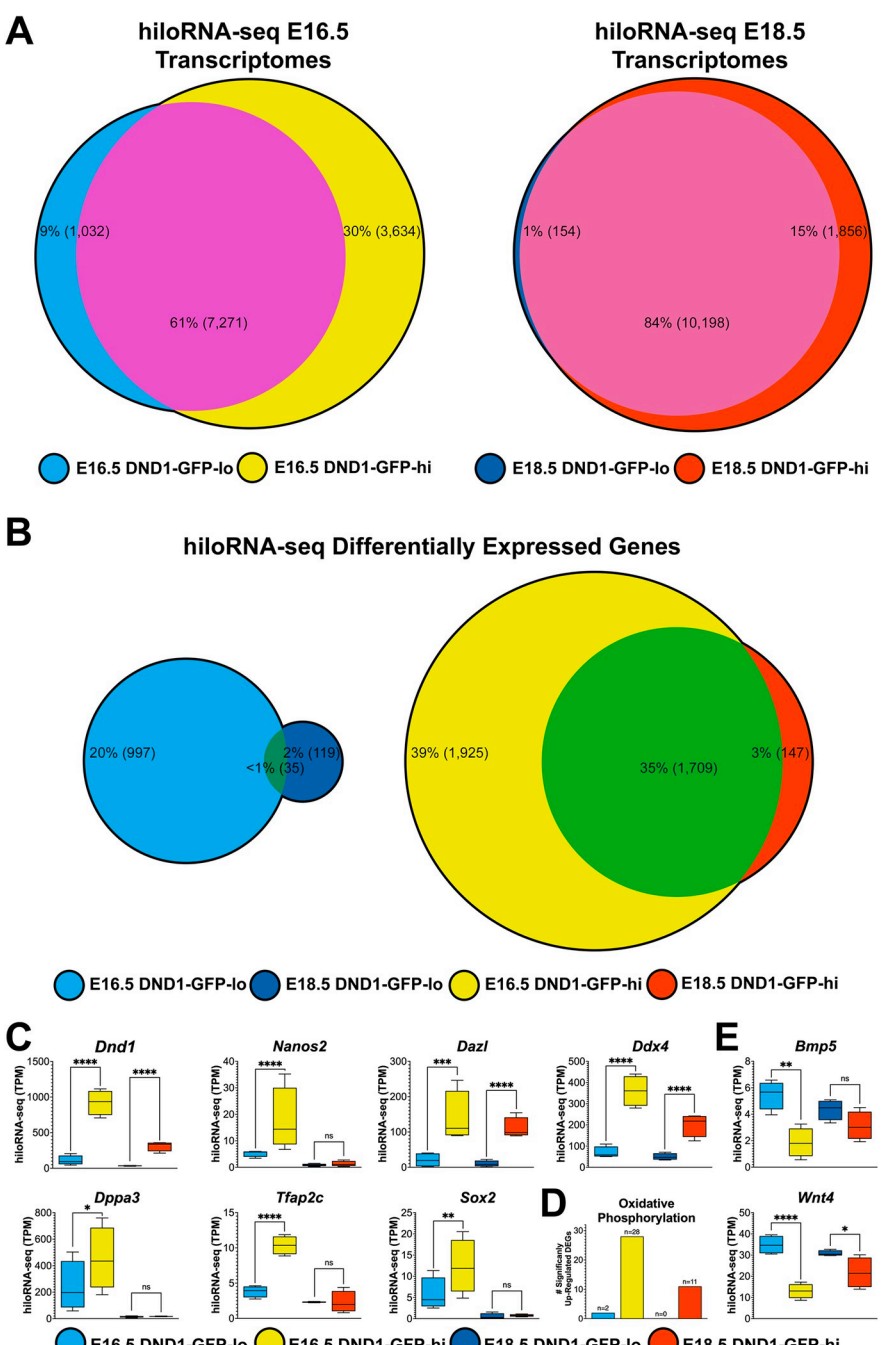

**Fig 2. Although the germ cell transcriptome is mostly shared between DND1-GFP-lo and DND1-GFP-hi cells at E16.5 and E18.5, the subset of significant differentially expressed genes indicate functional differences between DND1-GFP-lo and DND1-GFP-hi cells. A:** Global shared (pink) versus DND1-GFP-lo (blue) or DND1-GFP-hi (yellow; red) specific expression at E16.5 and E18.5 from hiloRNA-seq analysis. **B:** Euler plots of E16.5 DND1-GFP-lo (blue), E16.5 DND1-GFP-hi (yellow), E18.5 DND1-GFP-lo (dark blue), E18.5 DND1-GFP-hi (red) show global shared and significant differentially expressed genes. **C:** Expression of *Dnd1* declined between E16.5 and E18.5 but was consistently higher in Dnd1-GFP-hi cells. Other markers associated with the germline stem cell population behaved in a similar manner. **D:** Transcripts of genes associated with the oxidative phosphorylation pathway were elevated in DND1-GFP-hi cells. **E:** Examples of intercellular signaling genes that were expressed higher in Dnd1-GFP-lo cells. hiloRNA-seq expression P-value (DESeq2) between DND1-GFP-lo and DND1-GFP-hi cells at E16.5 and E18.5: not significant (ns), <0.05 (*), <0.01 (**), <0.001 (***), <0.0001 (****).

## DND1-GFP-lo and DND1-GFP-hi cells show differences in presence of phosphatidylserine on their cell surface

Several characteristics have been reported to distinguish populations of MGCs at E14.5 [30, 32]. We used immunofluorescent staining and confocal imaging to qualitatively correlate expression of a few of these markers with high or low expression of DND1-GFP at E14.5 (S5A–S5C Fig) and E18.5 (S5D–S5F Fig). For example, expression of the Line1 translation product, ORF1p, was associated with MGCs on a male differentiation trajectory [30]. However, ORF1p did not correlate with high or low levels of DND1-GFP (S5A and S5D Fig). Based on the metabolic differences we detected at the transcriptome level, we tested whether DND1-GFP-lo and DND1-GFP-hi cells showed higher levels of mitochondria using an antibody against TOMM20, but we found no clear differences between populations (S5B and S5E Fig), and levels of cPARP did not consistently correlate with high or low levels of DND1 at E14.5 or E18.5 (S5C and S5F Fig).

Gene ontology (GO) analysis of significantly differentially expressed genes that were over-expressed in either DND1-GFP-lo or DND1-GFP-hi cells identified apoptosis and cell death related genes in both cell populations (S3 Table). However, while DND1-GFP-hi cells favored genes related to the "Intrinsic Apoptotic Signaling Pathway" (S3 Table), DND1-GFP-lo cells favored genes related to the "Extrinsic Apoptotic Signaling Pathway" (S3 Table). To explore differences in cell death mechanisms between the two populations, we examined if DND1-GFP-lo cells had an increase in phosphatidylserine on the outer plasma membrane. Phosphatidylserine was shown to mark spermatogonia for phagocytosis by Sertoli cells in the adult testis [33–35]. To detect this membrane marker, we incubated dissociated E16.5 testes cells with a AF647-fluorescent conjugated antibody against Annexin V, which has high affinity for phosphatidylserine, and performed FACS analysis for Annexin V and DND1-GFP. DND1-GFP-lo cells, which dramatically over-express *Anxa5* only at E16.5, exhibit a significantly higher ($p<0.0001$) geometric mean of Annexin V fluorescence (Fig 3A and 3B). By comparing DND1-GFP-lo and DND1-GFP-hi cells to Annexin V signal (Fig 3C), we found that 70.0% of DND1-GFP-lo cells were Annexin V-positive, whereas only 3.5% of DND1-GFP-hi cells were Annexin V-positive (Fig 3D, $p = 0.0014$). Moreover, DND1-GFP-lo cells represented a mean of 99.25% of all Annexin V positive cells and just 37% of all Annexin V-negative cells (Fig 3D, $p = 0.0008$).

## Global analysis revealed that DND1-GFP-hi cells express higher levels of chaperones, chromatin/ histone regulators, and genes associated with translation

To determine whether transcripts that differed between DND1-GFP-lo and DND1-GFP-hi cells belonged to specific categories or protein classes, significantly over-expressed gene lists were generated by stage and DND1 expression level (up in: E16.5 DND1-GFP-lo, E16.5 DND1-GFP-hi, E18.5 DND1-GFP-lo, or E18.5 DND1-GFP-hi). Over-expression gene lists were submitted to PANTHER for grouping by protein class [36, 37]. This approach resolved signatures that were strikingly different between DND1-GFP-lo and DND1-GFP-hi cells at both time points (Fig 4A). PANTHER revealed that DND1-GFP-lo cells showed higher levels of genes associated with defense/immunity (*Skint7*, *Skint8*, *Hsh2d*, *Cle18a*), intercellular signaling (*Bmp5*, *Tgfa*, *Wnt4*, *Fgf10*), transmembrane signal receptors (*Amhr2*, *Igflr1*, *Ddr2*, and *Prlr*) and transporters (*Rhag*, *Catsper3*, *Kcnmb4*, Slc's) at both timepoints (Figs 4A, 2E and S6). In contrast, transcripts encoding chaperones, chromatin binding/regulation, nucleic acid metabolism, and translational proteins were elevated in DND1-GFP-hi cells at both timepoints (Figs 4A, S7 and 5A–5C).

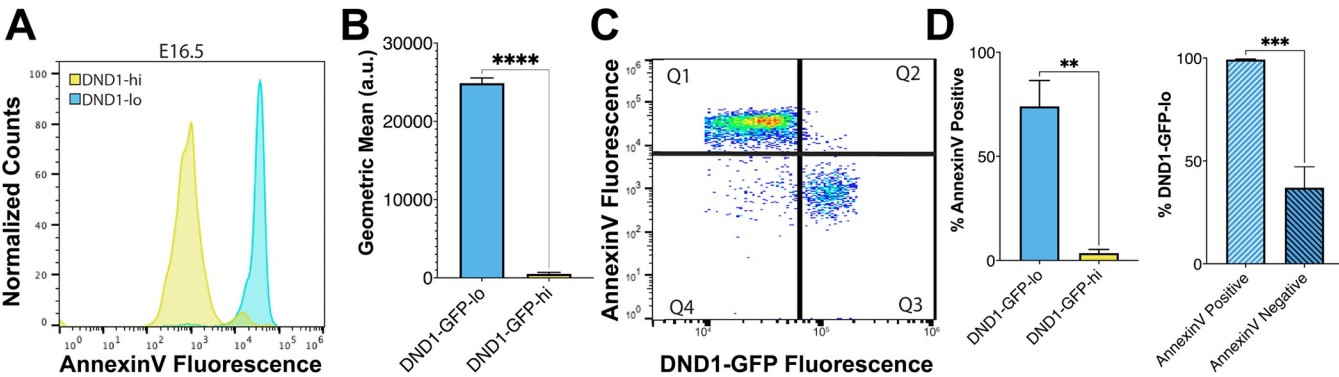

**Fig 3. DND1-GFP-lo cells express more phosphatidylserine on their outer membrane, as seen through Annexin V fluorescence. A:** Representative profiles of Annexin V fluorescence in E16.5 DND1-GFP-hi (yellow) and E16.5 DND1-GFP-lo (sky blue) populations. **B:** Geometric mean of Annexin V fluorescence in DND1-GFP-lo and DND1-GFP-hi cells. **C:** Representative fluorescence profile of DND1-GFP and Annexin V. Q1 (quadrant 1) = DND1-GFP-lo cells + Annexin V-positive. Q2 = DND1-GFP-hi cells + Annexin V-positive. Q3 = DND1-GFP-hi cells + Annexin V-negative. Q4 = DND1-GFP-lo cells + Annexin V-negative. **D:** Percent of DND1-GFP-lo cells and DND1-GFP-hi cells that are Annexin V-positive. **E:** Percent of total Annexin V-positive and negative cells that are DND1-GFP-lo cells. A-E: Technical replicates = 4, biological replicates > 2 fetal testes. P value: ****<0.0001, *** = 0.0008, ** = 0.0014.

In most cases, differences between DND1-GFP-lo and DND1-GFP-hi cells were more pronounced at E16.5 with expression levels converging at E18.5. We identified 757 transcripts significantly converged between DND1-GFP-lo and DND1-GFP-hi cells: 419 transcripts decreasing in DND1-GFP-hi cells and 338 transcripts increasing in DND1-GFP-hi cells to approach E18.5 DND1-GFP-lo levels. Based on GO analysis in g:Profiler decreasing transcripts were primarily associated with metabolic processes, whereas increasing transcripts were primarily associated with cell migration and signaling.

Despite the convergence of some transcripts, many epigenetic regulators maintain higher expression levels (TPM≥5.0, log2-fold change (L2FC)≥1.0, p-val<0.05) in DND1-GFP-hi cells throughout G0. This group included two members of the BET-BRD (bromo and extra terminal domain) family, two Dnmt (DNA methyltransferase) members, three Hdac (histone deacetylase) members, four repressive members of the Kdm family (lysine demethylase), four members of the Smarc (matrix-associated, actin dependent chromatin regulators) family, and five components of the SAGA complex (*Trrap*, *Kat2a*, *Taf5l*, *Taf6l*, *Taf9*), a highly conserved histone deubiquitinase and histone acetyltransferase complex. Genes involved in chromosome stability were also elevated in DND1-GFP-hi cells (Fig 4B). For example, four members of the Tudor family involved in transposon repression (Fig 4B) and *Piwil1*, *Piwil2*, and *Piwil4*, which mediate the piRNA pathway, showed higher expression in DND1-GFP-hi cells (Fig 5A).

In addition, genes encoding proteins involved in RNA metabolism and translation were expressed at higher levels in DND1-GFP-hi cells. The majority expressed at E18.5 were also expressed at E16.5 (Figs 4A and 5A–5C). We focused on transcripts that encoded genes classified as RNA binding proteins, ribosomal subunit genes, and Golgi vesicle transport that were highly significant at both E16.5 and E18.5. RNA binding proteins included *Mael*, *Piwil4*, *Rbfox2*, and an exportin (*Xpo5*) (Fig 5A). Many genes encoding ribosomal subunits (including Mrp and Rp family members) were over-expressed in DND1-GFP-hi cells (Fig 5B), as well as a large group of transcripts encoding proteins involved in Golgi vesicle transport (including *Exoc3*, *Pfdn1*, *Sys1*, and *Tmed9*) (Fig 5C).

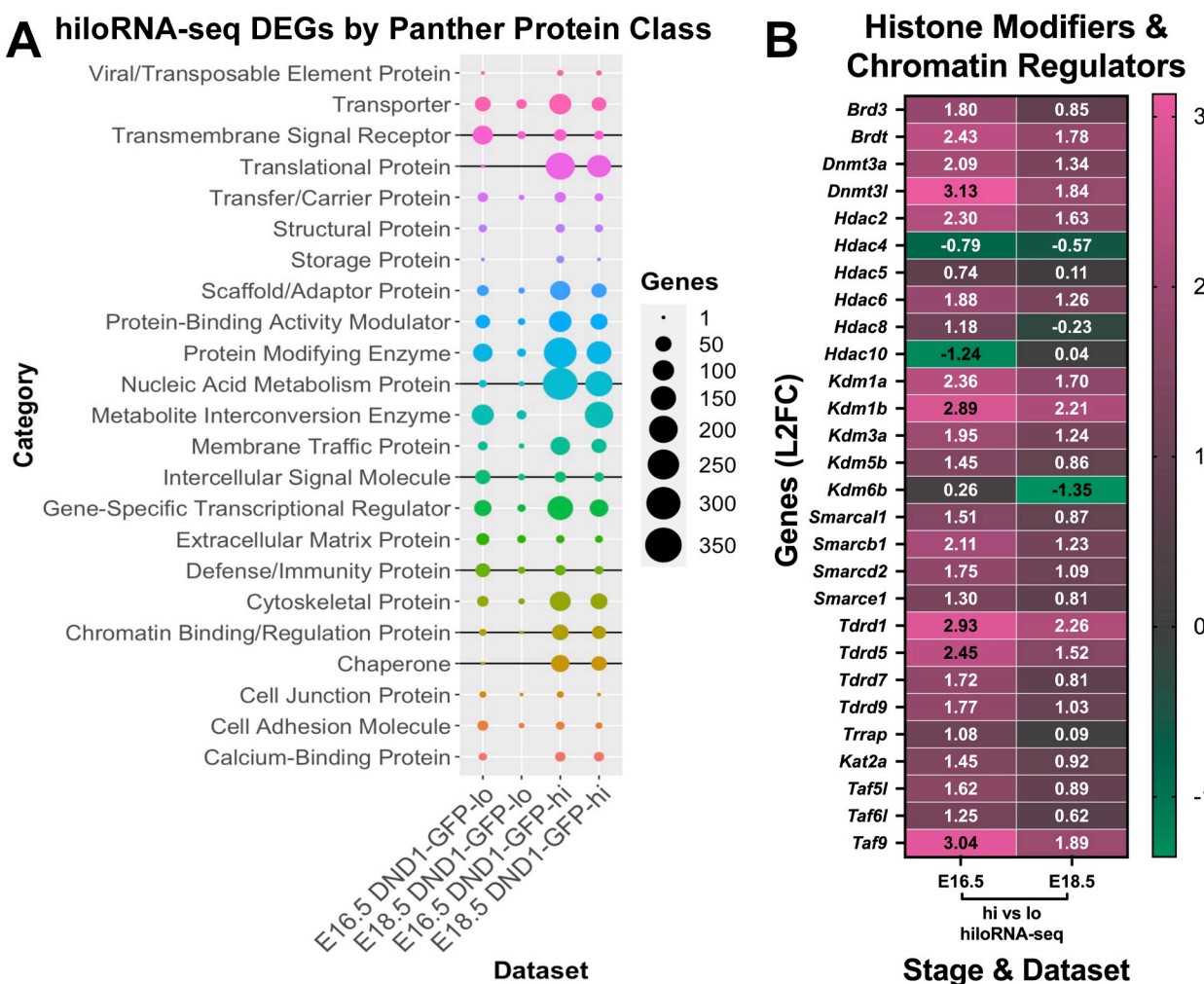

**Fig 4. Expression of some protein class categories are significantly different in DND1-GFP-hi vs DND1-GFP-lo cells and consistent between E16.5 and E18.5. A:** PANTHER analysis of significant differentially expressed genes in E16.5 and E18.5 DND1-GFP-lo cells, and E16.5 and E18.5 DND1-GFP-hi cells. Results displayed as protein class categories by dataset and graphed with the number of genes in each category corresponding to the size of the circle. Protein classes distinguishing DND1-GFP-lo cells at both stages include "Defense and Immunity", "Transmembrane Signal Receptor" and "Intercellular Signal Molecule" (noted with black vertical line). Protein classes distinguishing DND1-GFP-hi cells at both stages include "Chaperone", "Chromatin Binding/Regulation", "Nucleic Acid Metabolism", and "Translational" (noted with black vertical line). **B:** Heat map of chromatin / epigenetic regulators showing: L2FC in DND1-GFP-hi vs DND1-GFP-lo cells from hiloRNA-seq (E16.5, E18.5).

## RNA immunoprecipitation sequencing (RIP-seq) identified mRNA targets of DND1 at E14.5, E16.5, and E18.5

One goal of producing the $Dnd1^{GFP}$ knock-in line was to use the tagged protein for RNA immunoprecipitation sequencing (RIP-seq) analysis from germ cells in vivo. Previous results from analysis of a DND1 tagged allele in HEK293 cells, or in vitro with ESCs or derived germ-line stem cell lines identified targets of DND1 [2, 5, 22, 23, 38, 39]. However, these experiments missed in vivo targets expressed specifically in MGCs during the last third of fetal development and could not address the dynamics of the in vivo system. We recently used RIP-RT-qPCR in a proof-of-principle experiment to assay for a group of predicted targets of DND1 in vivo [27]. To extend this preliminary analysis, we isolated MGCs from 3 stages of development, E14.5 (the beginning of G0), E16.5 (a midpoint in G0), and E18.5 (just prior to birth), and performed RNA immunoprecipitation using an antibody against GFP [40–42]. After performing initial

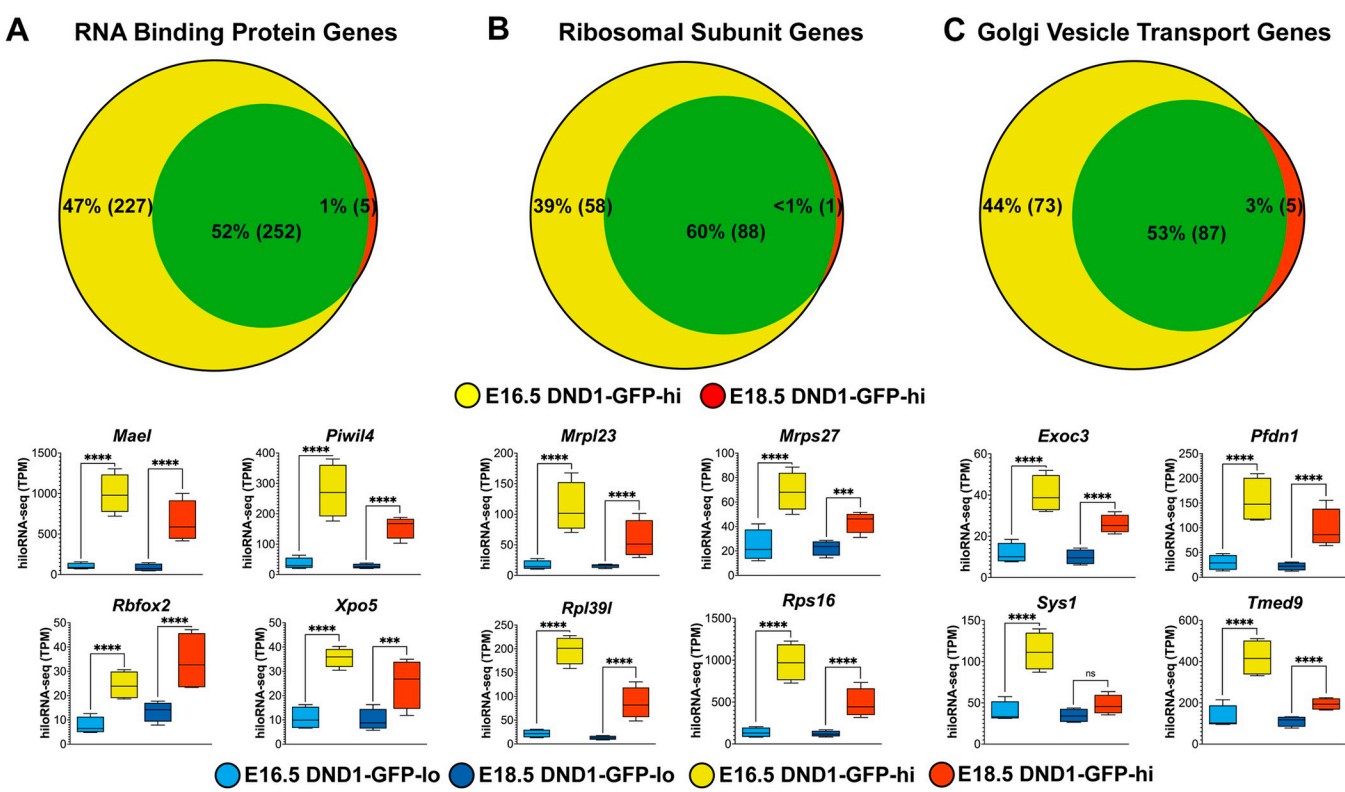

**Fig 5. In DND1-GFP-hi cells, most transcripts for proteins involved in RNA binding, ribosomal subunits, and Golgi vesicle transport are either shared between E16.5 and E18.5, or unique to E16.5.** Venn diagrams showing temporal distribution (E16.5 (yellow), E18.5 (red)) of genes in three GO biological processes significantly over-represented in DND1-GFP-hi cells compared to DND1-GFP-lo cells: (**A**) RNA binding protein genes, (**B**) ribosomal subunit genes, and (**C**) Golgi vesicle transport genes. Box-and-whisker plots show expression for example genes in each category. hiloRNA-seq expression P-value (DESeq2) between DND1-GFP-lo and DND1-GFP-hi cells at E16.5 and E18.5: not significant (ns), <0.05 (*), <0.01 (**), <0.001 (***), <0.0001 (****).

quality control checks and analysis (see Materials and Methods; S8 and S9A Figs) we used differential analysis between paired RNA input and immunoprecipitation (IP) samples to determine significant enrichment/non-enrichment of transcript binding targets of DND1 (S9B Fig and S4 Table). Quantification and comparison of the targets at each timepoint produced an Euler plot [43] showing that 17% of targets are conserved at all timepoints (E14.5, E16.5, E18.5) with only 13% unique to E14.5, 16% unique to E16.5, 48% unique to E18.5, and a limited group of targets (≤5%) shared by only two stages (Fig 6A). The full lists of targets at each timepoint (S2 Table) were submitted for gene ontology analysis in g:Profiler [44]. Out of the list of 363 biological processes collectively ascribed to target lists at each time point (S5 Table), 109 categories were present at all three timepoints (S5 Table).

Significant terms (adjusted p-value ≤0.01) were analyzed in a bubble plot depicting the number of DND1 targets (Y-axis) that were part of a given GO term at each stage (X-axis) with the significance (-log$_{10}$ adjusted p-value) represented by the size of the bubble. This analysis demonstrated that the number of target transcripts in a GO category often both increased over time and increased in significance score as a category (Fig 6B). Thematically many of these GO terms were related to regulation of cell differentiation through control of cell cycle or metabolism (Fig 6B, upper plot), and cell movement or intra/extra-cellular structure (Fig 6B, lower plot). Transcript targets in each of these three categories include *Bmp1* & *Pard3* (regulation of cell differentiation through cell cycle control), *Nox4* & *Atp2b4* (metabolism), *and Lmna* & *Mmp2* (cell movement or intra/extra-cellular structure) (Fig 6C). Many transcripts encoding

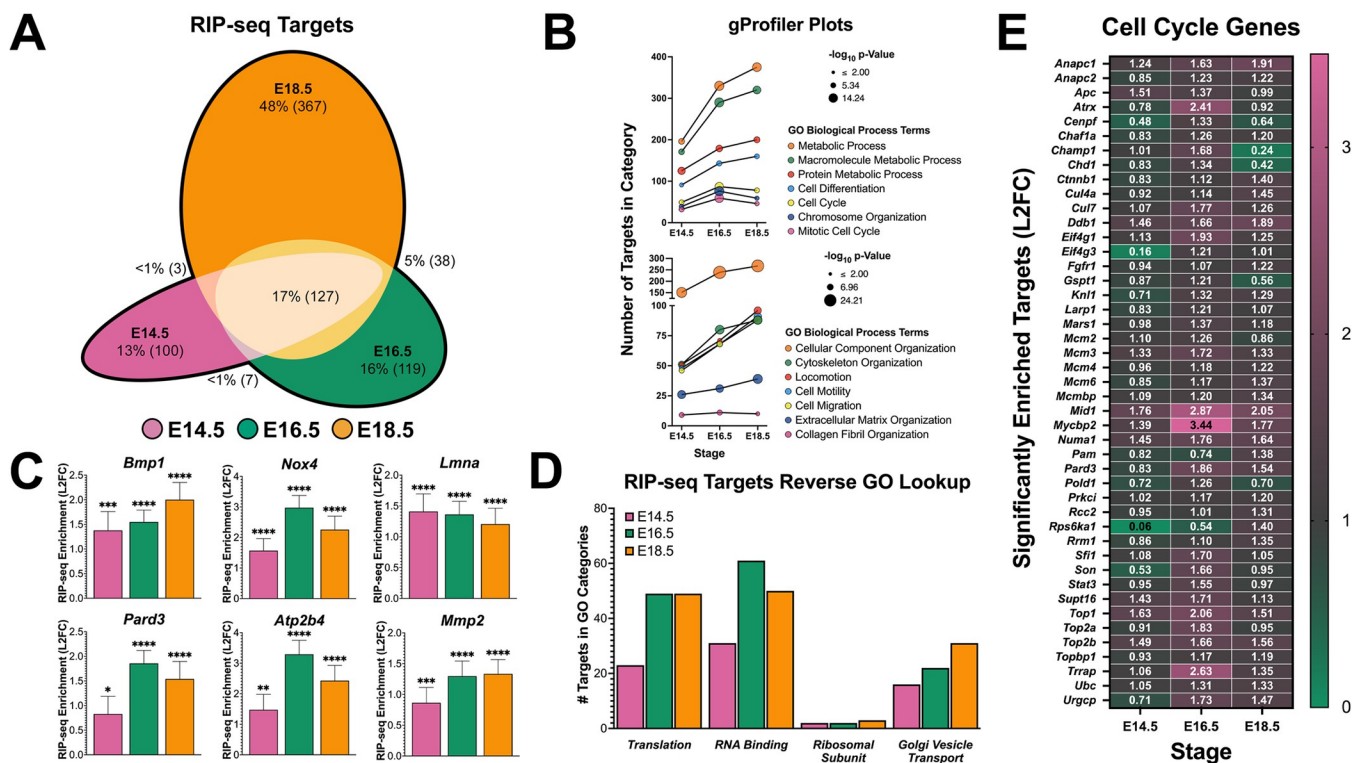

**Fig 6. RIP-seq reveals conservation of many DND1 targets at all 3 stages. A:** Euler plot of temporal distribution (E14.5 (pink), E16.5 (green), E18. (orange)) of significantly enriched transcript targets of DND1. **B:** Bubble plot from g:Profiler analysis of significantly enriched transcript targets of DND1 showing change in number of targets (Y-axis) for prominent GO biological processes related to cell cycle and metabolism (upper), and cell movement and ECM-cytoskeleton organization (lower) the -log10 P-value of each GO biological process is denoted by bubble size with developmental stages progressing along the X-axis. **C:** Bar graphs showing target enrichment as a function of L2FC at E14.5, E16.5, E18.5. RIP-seq enrichment P-value (DESeq2): not significant (ns), <0.05 (*), <0.01 (**), <0.001 (***), <0.0001 (****). **D:** number of significantly enriched transcript targets at E14.5, E16.5, E18.5 for GO biological processes identified in hiloRNA-seq analysis (Fig 4A): translation, RNA binding, ribosomal subunit, Golgi vesicle transport. **E:** Heatmap of significantly enriched transcript targets at E14.5, E16.5, E18.5 for the GO biological process cell cycle.

translational proteins, RNA binding proteins, ribosomal subunits, and Golgi vesicle transport were expressed at higher levels in DND1-GFP-hi cells (Fig 5A–5C). To investigate whether these same GO categories were over-represented among DND1 targets, the GO-derived gene lists were used in a SQL database to identify genes that were also targets of DND1. Transcripts for many genes associated with translation, RNA binding, and Golgi vesicle transport were targets of DND1, especially at E16.5 and E18.5. However, despite the abundance of ribosomal subunit genes differentially expressed between DND1-GFP-lo and DND1-GFP-hi cells, very few ribosomal subunit transcripts were targets of DND1 (Fig 6D). LaminA (*Lmna*) was one of a small number of transcripts that showed the most significant association with DND1 at E14.5 (Fig 6C). Transcripts associated with the cell cycle tended to show highest association with DND1 at E16.5, whereas transcripts associated with metabolic process, intracellular organization, and cell motility showed an increasing association with DND1 from E14.5 to E18.5 and were highly represented among the 367 targets that were only significant at E18.5.

Cell cycle regulation is critical in fetal male germ cell development. The resultant g:Profiler category of cell cycle contained a list of target genes sometimes with oppositional roles in the cell cycle (Fig 6E). This included genes involved with positive regulation of cell cycle (*Atrx*, *Numa1*, *Pold1*), negative regulation of cell cycle (*Cenpf*, *Ctnnb1*, *Knl1*), and the DNA damage checkpoint (*Cul4a*, *Ddb1*, *Topbp1*). Transcript targets in these three cell-cycle sub-categories did not present a global pattern of enrichment in any category. There were genes that were

targeted by DND1 only later in the time course (*Ctnnb1*, *Pam*, *Topbp1*), and others that peaked as targets at E16.5 (*Eif4g1*, *Atrx*, *Trrap*). The three strongest targets in this cell cycle sub-analysis (*Mid1*, *Mycbp2*, *Trrap)* were most significantly associated with DND1 at E16.5 (Fig 6E). *Mid1* (Midline 1), a member of the TRIM (tripartite motif) family, is an E3 uibiquitin ligase that regulates microtubule anchoring as part of cytokinesis [45]. *Mycbp2* (MYC Binding Protein 2) and *Trrap* (Transformation/Transcription Domain Associated Protein) are both MYC binding proteins that are linked to E2 activity and transcriptional activation [46–54].

## Many epigenetic regulators that are differentially expressed dependent on the level of DND1 are also targets of DND1

Of the 869 DND1 transcript targets identified in our analysis, 164 were found in E16.5 DND1-GFP-hi cells (18.87%), 81 in E18.5 DND1-GFP-hi cells (9.32%), 81 in E16.5 DND1-GFP-lo cells (9.32%), and 6 in E18.5 DND1-GFP-lo cells (0.01%). A major common class between hiloRNA-seq and RIP-seq were genes related to epigenetic regulation, which we previously found to be significantly disrupted in germ cells carrying the *Ter* mutation (*Dnd1^{Ter/Ter}*) [23]. We explored the overlap between epigenetic regulators that were differentially expressed in the hiloRNA-seq dataset (usually over-expressed in DND1-GFP-hi cells) (Fig 4B) and enriched as in vivo targets of DND1 at E14.5, E16.4 and E18.5 (L2FC>0.5, p-val<0.05) (S10A Fig). *Brd3* and *Brdt* are both over-expressed in DND1-GFP-hi cells, but only *Brd3* maps as a target. The DNA methyltransferases, *Dnmt3a* and *Dnmt3l*, are higher in DND1-GFP-hi cells, and both enrich as targets of DND1 at one or more stages (Figs 4B, S10A and S10B). *Hdac2* and *Hdac6* are over-expressed in DND1-GFP-hi cells, but only *Hdac4*, *Hdac5*, and *Hdac6* enrich as targets of DND1. Several H3K4 demethylases *Kdm1a*, *Kdm1b*, *Kdm3a*, *Kdm5b*, *and Kdm6b* enrich as targets of DND1, but only the first four of these, which are associated with repression, are higher in DND1-GFP-hi cells (Figs 4B, S10A and S10B). Although four of the SWI/SNF-related matrix-associated, actin dependent regulators (Smarcs) are higher in DND1-GFP-hi cells, only two of these (*Smarc1* and *Smarcd2*) enrich as targets of DND1 (S10A and S10B Fig). Four members of the Tdrd family of transposon regulators are higher in DND1-GFP-hi cells, and all four of these (*Tdrd1*, *Tdrd5*, *Tdrd7*, *Tdrd9*) are targets of DND1. Multiple members of the highly conserved histone deubiquitinase and histone acetyl-transferese SAGA complex (*Trrap*, *Kat2a*, *Taf5l*, *Taf6l*, and *Taf9*) are higher in DND1-GFP-hi cells, but only one central co-factor (*Trrap*) is a consistent target of DND1. As a further step, we expanded our SQL database to encompass our previously published *Dnd1^{Ter}*-seq dataset [23]. Nearly all of these chromatin regulators are genes that fail to be up-regulated in *DND1^{Ter/Ter}* mutant germ cells by E14.5 (S10A Fig). Only one histone deacetylase (*Hdac10*) is significantly upregulated in *DND1^{Ter/Ter}* mutant germ cells and, notably, *Hdac10* is not a target of DND1 and not significantly differentially expressed in E16.5 DND1 cells (S10A Fig). Similar three-way analysis was conducted to identify DND1 transcript targets, differentially expressed between DND1-GFP-hi and DND1-GFP-lo cells that were dysregulated in *Dnd1^{Ter/Ter}* mutant germ cells related to extracellular matrix organization and the cell cycle. Unlike epigenetic regulation, three-way analysis for extracellular matrix organization and cell cycle genes had less consistent regulatory patterns across all three datasets (S2 Table).

Two recent publications have reported groups of genes associated with the transition of pro-spermatogonia to spermatogonia identity [16, 55]. Part of this transitional process was reported to be characterized by expression of genes related to cell migration, a process that was strongly represented in the gene ontology analysis of our RIP-seq data. We repeated our two-way analysis with the family of collagens, many of which were strongly enriched as DND1 targets (S11A Fig), to identify any patterns in our hiloRNA-seq dataset (S11B Fig). Notably there

was very limited significant differential expression of these collagens between DND1-GFP-lo and DND1-GFP-hi cells at E16.5 or E18.5. Five out of six of the genes reported by the Wilkinson group to be associated with the intermediate pro-spermatogonia identity [55] showed higher expression in DND1-GFP-hi cells, and all six enriched as binding targets of DND1 (S11C Fig). Of the genes identified by the Zhao group [16] (*Zxdc*, *Tie1*, *Myog*, *Mef2c* and *Krt18*), *Zxdc*, *Myog* and *Krt18* are expressed at higher levels in E16.5 and E18.5 DND1-GFP-hi cells, while *Tie1* and *Mef2c* are expressed similarly between DND1-GFP-lo and hi cells. Notably, only *Zxdc* enriched as a target and only at E16.5 and E18.5 (S11D Fig).

Three Y-linked genes (*Eif2s3y*, *Uty*, *Smcy*) were significantly differentially expressed between DND1-GFP-lo and DND1-GFP-hi cells at E16.5. The Y-linked translation initiation factor, *Eif2s3y*, which is known to be important in spermatogonial proliferation and differentiation [56, 57], was elevated in DND1-GFP-hi cells at E16.5, while both Y-linked epigenetic regulators, *Uty* and *Smcy*, were elevated in DND1-GFP-lo cells at E16.5 and E18.5. We investigated the expression of the X-linked homologs (*Eif2s3x*, *Utx*, *Smcx*) in DND1-GFP-hi and DND1-GFP-lo cells and if any of these sex chromosome genes enriched as targets. None of the X-linked genes (*Eif2s3x*, *Utx*, *Smcx*) were significantly differentially expressed between DND1-GFP-lo and DND1-GFP-hi cells. All the epigenetic regulators enriched as targets at least at one stage, while *Uty*, *Smcx*, and *Smcy* enriched as targets at all stages (*Utx*, at E14.5 only) (S12 Fig).

## Discussion

We show that there are two populations of MGCs expressing low and high levels of DND1 during G0 arrest in late fetal life. These two populations were first discovered by FACS analysis of MGCs expressing the DND1 allele tagged with GFP at the endogenous locus (*Dnd1^GFP^*). These findings were supported by live imaging, quantitative microscopy, and analysis of single cell sequencing datasets from 2 independent labs, which also reveal MGC populations expressing high and low levels of native *Dnd1* [31, 32]. DND1-GFP-lo and DND1-GFP-hi populations do not correlate with any markers of MGC heterogeneity previously reported [30]. However, our analyses revealed that the transcriptomes of DND1-GFP-lo cells are highly correlated between E16.5 and E18.5, and significantly different from DND1-GFP-hi cells (which are also highly correlated between both stages). The finding that higher levels of phosphatidylserine on the surface of E16.5 MGCs (as determined by Annexin V detection) is correlated with low levels of DND1 suggests that DND1-GFP-hi cells may be favored to survive and give rise to the pro-spermatogonial population. Prominent among genes that show higher expression in DND1-GFP-hi cells, and also enrich as targets of DND1, is a group of genes encoding epigenetic regulators and translational machinery. Some of these novel direct targets have been previously shown to be dysregulated in the presence of mutated *Dnd1* (*Dnd1^Ter/Ter^*) [23], while other targets identified by us (*Tdrd7*, *Rock2*, *Mylk*, *Anxa5*) have previously been reported as targets of DND in zebrafish [24, 38]. These results suggest that DND1 coordinates RNA regulons of related epigenetic factors that control reprogramming of MGCs during cell cycle arrest.

Defects in RNA metabolism are likely to underlie teratoma formation in the presence of the *Dnd1^Ter/Ter^* mutation. The eukaryotic translation initiation factor, *Eif2s2*, has been shown to interact genetically with DND1. Although full deficiency for *Eif2s2* was embryonic lethal, partial deficiency protected against teratomas by restricting germ cell proliferation and differentiation [58, 59]. Several EIFs enrich as targets of DND1 (*Eif3a*, *Eif4g3*, *Eif3c*). Other EIFs were significantly differentially expressed between DND1-GFP-lo and DND1-GFP-hi cells (*Eif3c*, *Eif3f*, *Eif2s2*, *Eif2s3y*). *Eif3f* has been shown to interact with DND1 in the context of Xenopus germline development [60]. *Eif2s3y* has been shown to be important in spermatogonial

differentiation and proliferation after birth [56, 57]. There is also Increasing evidence suggesting that ribosomal subunits have specificity [61]. It will be very interesting to determine whether certain of the ribosomal subunits expressed at high levels in DND1-GFP-hi cells have specific translational roles in MGCs. Overall, the interaction of DND1 with these and other components of eukaryotic translational machinery needs further study.

Consistent with the idea that DND1-GFP-hi cells are a select population of MGCs, at E16.5, they express much higher levels of genes associated with the germline stem cell population including *Dazl*, *Ddx4*, *Nanos2*, *Dppa3*, *Tfap2c*, *Pou5f1*, and *Sox2*, although transcripts for all these genes decline by E18.5. DND1-GFP-hi cells show higher expression of genes associated with oxidative phosphorylation, and also express elevated levels of other RNA binding proteins relevant to male fate specification such as *Dazl* and *Nanos2*. DAZL is associated with the transition to MGC differentiation [62] and NANOS2 is well documented to work in complex with DND1 to target transcripts of pluripotency genes and activating cyclins for degradation [1, 25, 63]. *Dazl* has also been shown to be targeted by NANOS2 as part of negative regulation and commitment to MGC fate [64].

DND1-GFP-hi cells express higher levels of epigenetic regulators. It is well established that MGCs undergo extensive epigenetic reprogramming during G0 arrest [18–21]. Among the genes elevated in DND1-GFP-hi cells were four members of the Tdrd family of transposon regulators, several DNA methyltransferases, four repressive chromatin regulators of the Kdm family, four members of the SWI/SNF-related Smarc family, and multiple members of the histone deubiquitinase and histone acetyltransferese SAGA complex. Many of these gene transcripts are also targets of DND1, suggesting that DND1 regulates their availability during cell cycle arrest. The regulation of multiple epigenetic enzymes by DND1 suggests that one reason that *Dnd1^{Ter/Ter}* mutant germ cells tend to give rise to teratomas is a failure to epigenetically silence somatic gene expression.

MGCs undergo hypertranscription from the stage when they begin to populate the gonad until at least E15.5 [65]. Hypertranscription in MGCs is regulated by MYC proteins that trigger release of paused RNA Pol II [65]. Interestingly, the Pol II complex is activated by the addition of BRD4 [66, 67], which is expressed at higher levels in DND1-GFP-hi cells and maps as a target of DND1 at E16.5 and E18.5. DND1 could also function upstream of MYC proteins and other transcription factors. *Trrap*, a highly conserved and well documented regulator of MYC and other transcription factors, is a target of DND1 that peaks at E16.5 (Figs 6E, S10A and 10B) [46–53]. It is possible that hypertranscription itself is under partial DND1 control.

The large number of RBPs present in MGCs may be essential for the post-transcriptional regulation of the many transcripts that accumulate in MGCs as a result of hypertranscription [65]. A novel finding in this study was the large group of transcripts encoding proteins associated with translation that are up-regulated in DND1-GFP-hi cells and are also bound by the DND1 protein. DND1 binds many transcripts related to vesicle fusion and Golgi function, and many of these transcripts are also up-regulated in DND1-GFP-hi cells. These data suggest that DND1 not only regulates transcript stability but may also control when or where transcripts are translated, acting as a hub for post-transcriptional regulation in the germline as suggested by Gross-Thebing and Raz [68]. It is unclear whether or how transcripts bound by DND1 and sequestered in various germ granules are transferred to the ribosome for translation.

Results from this study raise many questions. First, what is responsible for low or high levels of DND1? Since this study began, we have seen variations in the proportion of DND1-GFP-lo and DND1-GFP-hi cells based on FACS analysis. Are DND1 levels in MGCs regulated by surrounding somatic cells or are levels of DND1 intrinsically regulated? TFAP2c (which is highly upregulated in DND1-GFP-hi cells) was reported to regulate DND1 levels in induced primordial germ cells (PGCs) [69]. TFAP2c responds to α-ketogluterate levels, which suggests a

metabolic connection. However, we found no correlation with levels of mitochondria, which we used as a surrogate or energy differences in DND1-GFP-lo and DND1-GFP-hi cells.

Second, acting in its role as a RNA-binding protein, DND1 is reported to stabilize transcripts in some instances [5, 24] but has also been shown to destabilize transcripts through cooperation with NANOS2 as part of the CNOT complex to bring transcripts to the processing (P) bodies for degradation [1, 2, 70]. The diverse group of DND1 targets reported here, including positive regulators of the cell cycle, which we predict are sequestered and degraded by DND1 during G0, as well as epigenetic enzymes, which we predict are protected by DND1, is consistent with more than one role for DND1. It is possible that DND1 acts to degrade transcripts at one stage, but to protect transcripts at another stage. However, the failure to find a consistent pattern in transcripts bound by DND1 at early or late stages suggests that it can perform either function at all stages. It is unclear whether this depends on characteristics of the target, its cellular localization, and/or other interacting proteins such as NANOS2 [1, 2, 70]. Cross examination of our RIP-seq and hilo-RNA-seq datasets revealed a limited number of DND1 targets that were significantly up-regulated or down-regulated from E16.5 to E18.5 specifically in DND1-GFP-lo or DND1-GFP-hi cells. Certain of these transcripts could be stabilized or destabilized in a DND1-GFP-lo or DND1-GFP-hi specific manner. However up-regulation could also be the product of hypertranscription occurring during this period. Future molecular follow up is needed to untangle expression changes due to transcriptional vs. post-transcriptional regulation.

Third, we have not yet determined whether populations of DND1-GFP-lo and DND1-GFP-hi cells are episodic or clonal. While DND1-GFP-hi cells are sometimes found in clusters within cords (as might be expected of clones), sometimes they are highly dispersed. In theory, episodic levels of DND1 could be compatible with sequestration of transcripts in high cells, followed by release for translation in low cells. Information about whether translational differences exist between the two populations is the next goal.

The biological significance of DND1-GFP-hi and DND1-GFP-lo cells is not yet entirely clear. A large proportion of MGCs are lost by the time of birth [30, 71, 72]. Consistent with this, both DND1-GFP-hi and DND1-GFP-lo cells express genes related to cell death and apoptosis. Although mechanism(s) regulating MGC apoptosis or protection from apoptosis during G0 are largely unexplored, our data suggest a preference for intrinsic apoptosis in G0 DND1-GFP-hi cells and extrinsic apoptosis in G0 DND1-GFP-lo cells. This is consistent with our findings of elevated phosphatidylserine in 16.5 DND1-GFP-lo cells, presumably, to mark cells for phagocytosis. Overall, these data suggest that MGCs expressing low levels of DND1 are more likely to be eliminated through an extrinsic cell death pathway that is independent of canonical apoptotic machinery, such as cPARP. Such germ cell death (GCD) mechanisms have been described in the *Drosophila* testes [73, 74].

One attractive possibility is that DND1-GFP-hi cells acquire the epigenetic modifications that facilitate differentiation as pro-spermatogonia, consistent with the accumulation of transcripts for epigenetic regulators in DND1-GFP-hi cells. In support of the idea that DND1-GFP-hi cells are a leading population of pro-spermatogonia, at E16.5 and E18.5, some markers recently reported to define a novel intermediate pro-spermatogonia identity are expressed at higher levels in DND1-GFP-hi cells [16, 55], but the pattern is inconsistent. Future work will aim to resolve these questions.

## Materials and methods

### Ethics statement

All mice were housed in accordance with National Institutes of Health (NIH) guidelines, and experiments were conducted with the approval of the Duke University Medical Center (DUMC) Institutional Animal Care and Use Committee (IACUC # A126-17-05).

## Colony maintenance and timed matings

The *Dnd1*<sup>EGFP</sup> knock-in allele was produced in the Duke Transgenic Core by CRISPR/Cas9 pronuclear injection into B6SJLF1/J (JAX stock# 100012) embryos as described [27]. Knock-in founders were bred with CD1 mice (Charles River Labs strain code 022). Progeny heterozygous for the knock-in were intercrossed to produce homozygous offspring, which were intercrossed to maintain a homozygous hybrid colony (*Dnd1*<sup>EGFP/EGFP</sup>, hereafter written simply as *Dnd1*<sup>GFP</sup>). Approximately every six generations, homozygous males were outbred with CD1 females. New heterozygous progeny were selected then intercrossed to restore a homozygous hybrid strain colony. For timed matings, homozygotes were intercrossed, females were checked for plugs, and staged as day E0.5 if positive.

## Fresh unfixed and unstained whole mount imaging

Fetuses from timed matings were dissected at E16.5 and E18.5 to collect testes. After excision, testes were washed in PBS followed by mounting on a slide (Thermo Fisher Scientific, cat# 12-550-15) in polyvinyl alcohol (PVA) with a coverslip (VWR, cat# 48366–227). Mounted samples were directly imaged using a SP8 Leica DM6000CS upright confocal microscope with affiliated Leica software (LAS AF 3). All tile stitching was done with Leica software, remaining image processing was done with FIJI (version 2.3.0/1.53f) [75] and Adobe Photoshop (version 23.1.0).

## Whole mount immunofluorescence on fixed samples

Whole mount immunofluorescence was performed for stages E14.5 and E18.5 as described [27, 76, 77]. Briefly, for E14.5 and E18.5, embryonic gonads were dissected from embryos and fixed in 4% paraformaldehyde in PBS at 4˚C overnight on a rocking platform. All samples were washed in PBS for 30–60 minutes, rocked for 15 minutes at room temperature at each step of a dehydration gradient (25%, 50%, 75% and 100% methanol in PBS), then stored in 100% fresh methanol at -20˚C. Prior to immunofluorescent staining, all samples were incubated at room temperature while rocking for 15 minutes at each step of the rehydration gradient (75%, 50% and 25% methanol in PBS), and washed twice in PBS for 15 minutes each. Samples were incubated rocking at room temperature in a permeabilizing solution (2% Triton X-100 in PBS) for 1 hour followed by incubation in blocking solution for 1 hour, then transferred to 4˚C in blocking solution (1% Triton X-100, 3% BSA, and 10% Horse Serum in PBS) with primary antibodies overnight. The following day, samples were washed by rocking in wash solution (1% Triton X-100 in PBS) three times at room temperature for 30 minutes each, then incubated at 4˚C in blocking solution with secondary antibodies and Hoechst (Invitrogen, cat# H3570) overnight. All antibodies are listed in Table 1. On day 3, samples were washed three times in wash solution rocking at room temperature for 30 minutes each, mounted in PVA (aka mowiol) mounting medium, and cover-slipped for imaging on a SP8 Leica DM6000CS upright confocal microscope with affiliated Leica software (LAS AF 3). All tile stitching was done with Leica software, remaining image processing was done with FIJI (version 2.3.0/1.53f) [75] and Adobe Photoshop (version 23.1.0).

## Quantification of DND1-GFP levels

The method for quantifying DND1-GFP-hi and DND1-GFP-lo cells (S2 Fig) was adapted from a previous publication [80]. Testes were fixed as previously described (refer to "Whole Mount Immunofluorescence on fixed samples" section). In fixed whole mount testes, endogenous GFP signal and Hoechst (Invitrogen, cat# H3570) staining were used for quantification

**Table 1. Antibodies.**

| Against | Raised In | Concentration | Vendor | Catalog # | Reference |
|---------|-----------|---------------|--------|-----------|-----------|
| GFP | Chicken | 1:5000 | Abcam | ab13970 | [27] |
| GCNA | Rat | 1:1000 | Abcam | ab82527 | [78] |
| ORF1p | Rabbit | 1:100 | Abcam | ab216324 | [30] |
| TOMM20 | Rabbit | 1:1000 | Abcam | ab186734 | [79] |
| cPARP | Rabbit | 1:100 | Cell Signaling Technology | 9544 | [30] |
| **Against** | **Raised In** | **Concentration** | **Vendor** | **Catalog #** | **Conjugate** |
| Chicken | Donkey | 1:1000 | Jackson Immuno Research | 703-545-155 | AF488 |
| Rat | Donkey | 1:1000 | Jackson Immuno Research | 712-165-153 | Cy3 |
| Rabbit | Donkey | 1:1000 | Jackson Immuno Research | 711-605-152 | AF647 |
| Annexin V | N/A | 1drop/200μL | Invitrogen | R37175 | AF647 |

in each experiment. A total of 50 individual MGCs over two biological replicates were randomly selected, and using the GFP signal, were individually cropped along the X and Y planes with FIJI [75]. Using the GFP signal and the Hoechst signal as visual guides, a projected z-stack was created for each MGC using the sum of intensities for the z-slices containing only the selected MGC. After manually creating a mask to outline the selected cell, DND1 fluorescent intensity for each MGC was obtained using GFP fluorescence mean intensity. Next, to account for imaging heterogeneity throughout the tissue, DND1-GFP fluorescent intensities were normalized to the average of the mean intensity of Hoechst staining, which is assumed to be equal across all MGCs. Because DND1-GFP-hi cells represent ~12% of the population in E16.5 MGCs and 50 MGCs were analyzed, we determined that the cluster of 5 cells (12% of 50 cells is approximately 5 cells) with the highest expression were among the DND1-GFP-hi population.

## scRNA-seq re-analysis

Demultiplexed fastq files of E16.5 male germ cell scRNA-seq data originally published in Law et al. 2019 (GSE124904) were downloaded to the Duke Computing Cluster (DCC) from the Sequence Read Archive (SRA) using the SRA toolkit (n = 3: SRR8427519, SRR8427520, SRR8427521) [31, 81]. SRA-based names for I1, R1, and R2 fastq files were changed to be compatible with Cell Ranger [82]. The Cell Ranger Count pipeline (read alignment, UMI counting) was performed on each biological replicate referenced to the mm10 mouse genome (GRCm38, v3.0.0). The Cell Ranger Aggregation pipeline was performed on the molecule_h5 files for each biological replicate to generate a single integrated E16.5 dataset for downstream analysis. Quality control and clustering was performed on the aggregated data using Seurat [83]. Stability indexing with Clustree [84] was used to choose a resolution of 0.1. The resultant clustered data was used for analysis and visualization in Seurat and exported to Loupe Browser (10x Genomics) for exploratory visualization.

## RNA Immunoprecipitation (RIP) and RNA Extraction

RNA immunoprecipitation (RIP) experiments were performed as previously described with adjustments to account for the very low input from fetal testes [27, 41, 42]. Briefly, fetal testes at either E14.5, E16.5, or E18.5 were dissected away from the mesonephros and collected. Pooled testes (6–16 testes total for a given biological replicate) were washed in DEPC PBS and incubated at 37°C in fresh clean Gibco TrypLE (Thermo Fisher Scientific, catalog #12563029) for 7 or 10 minutes depending on stage (E14.5, or E16.5 and E18.5 respectively). TrypLE was

inactivated with ice cold DEPC PBS and aspirated off testes. Testes were then resuspended in ice cold DEPC PBS and pipetted vigorously to dissociate cells. Cells were pipetted through a 0.32μm cell strainer into a fresh Eppendorf tube and pelleted by 4°C centrifugation at 1,000 x g for 10 minutes. The cell pellet was resuspended in 20μL complete Poly Lysis Buffer (PLB) [41], incubated on ice for 5 minutes and stored at -80°C for at least 24 hours. The PLB cell suspension was then thawed on ice and cellular debris was pelleted by 4°C centrifugation at 21,000 x g for 10 minutes. After centrifugation 3μL of supernatant was collected for RNA extraction as input sample. The remaining 17μL of supernatant was used for immunoprecipitation via incubation at 4°C on a tube rotator in NT2 Buffer [41] with NT2 Buffer washed GFP-Trap magnetic agarose beads (ChromoTek, catalog #gtma-20) for 2–4 hours. After immunoprecipitation, beads were pelleted, washed in NT2 buffer and pelleted again for RNA extraction as immunoprecipitation (IP) sample. RNA from input and IP samples was extracted using the RNeasy Micro Kit per the manufacturers protocol (Qiagen, catalog #74004). Extracted RNA from input and IP samples at E14.5, E16.5, and E18.5 was used in RNA-sequencing.

### RIP-seq analysis

After RNA immunoprecipitation and RNA extraction, paired input and immunoprecipitation (IP) samples were submitted for sequencing. Paired biological replicates per time point were sent for quality control (E14.5, n = 5 biological replicate pairs, E16.5, n = 4 biological replicate pairs, and E18.5, n = 5 biological replicate pairs; total of 28 samples). RNA quality control was measured on a Qubit (Thermo Fisher Scientific). All samples passed quality control and the 3 biological replicate pairs per time point with the lowest contaminant score and the most RNA were selected for library preparation. Sample libraries were prepared with SMART-Seq v4 Ultra Low Input RNA Kit (Takara Clontech Kit Cat# 63488). Libraries were sequenced on a NoveSeq 6000 (Illumina) as 50bp paired-end reads (~46 M reads/sample input, ~50 M reads/sample IP). Reads were mapped to the mm10 (GRCm38) mouse genome using Salmon with default settings (~78% mapping input, ~57% mapping IP) [85]. Mapped reads were annotated using the Ensemble *Mus musculus* reference (version 79). Read abundance from Salmon (TPM) values were used in post-annotation quality checks: correlation matrix, principal component analysis (PCA), scatter plots. Paired input and IP samples were then evaluated for transcript enrichment in IP vs input using DeSeq2 [86]. The resulting data table was used for refining target transcript gene lists in SQL [87, 88].

### Fluorescence Associated Cell Sorting (FACS)

Sorting was performed at the Duke Flow Cytometry Shared Resource core on a B-C Astrios Sorter. Initially, E18.5 DND1[GFP] testis samples compared with ovaries from female siblings used as negative controls, defined a gating protocol to distinguish between GFP negative, GFP positive-low, and GFP positive-high gonadal cells. The established gating protocol was used for all sorting experiments (E14.5, E16.5, E18.5, P3). Sorting was performed as described previously [27]. Briefly, stage E16.5 and E18.5 testes were collected as described in the RIP-RNA Extraction method section. However, after TrypLE treatment testes were dissociated in 3% BSA in DEPC PBS. After dissociation the cell suspension was passed through a 0.32μm cell strainer. Collected cells were then sorted on presence and absence of GFP (GFP positive and GFP negative) and level of GFP (GFP positive-low, and GFP positive-high). Cells were sorted directly into RLT buffer from RNeasy Micro Kit (Qiagen, catalog #74004). RNA extraction was performed as described in the RIP-RNA extraction method section.

## hiloRNA-seq analysis

Sorts were performed as described in the Fluorescence Associated Cell Sorting (FACS) section. After FACS and RNA extraction, paired DND1-GFP-lo and DND1-GFP-hi samples were submitted for sequencing. Four independently collected paired biological replicates per time point were sent for quality control (4 DND1-GFP-hi and 4 DND1-GFP-lo paired samples, for both E16.5 and E18.5, 16 samples total). RNA quality control was measured on a TapeStation (Agilent Technologies) using High Sensitivity RNA ScreenTape (Agilent Technologies). All samples passed quality control. Sample libraries were prepared with SMART-Seq v4 Ultra Low Input RNA Kit (Takara Clontech Kit Cat# 63488). Libraries were sequenced on a NoveSeq 6000 (Illumina) as 50bp paired-end reads (~54 M reads/sample DND1-GFP-hi, ~50 M reads/ sample DND1-GFP-lo). Reads were mapped to the mm10 (GRCm38) mouse genome using Salmon with default settings (~46% mapped DND1-GFP-hi, ~28% DND1-GFP-lo) [85]. Mapped reads were annotated using the Ensemble *Mus musculus* reference (version 79). Read abundance from Salmon (TPM) values were used in post-annotation quality checks: correlation matrix, principal component analysis (PCA). Paired DND1-GFP-lo and DND1-GFP-hi samples were then evaluated for significantly differentially expressed genes using DeSeq2 [86]. The resulting data table was used for refining gene lists in SQL [87, 88].

## Super-series (hiloRNA-seq, RIP-seq, $Dnd1^{Ter}$-seq) analysis

The resulting data tables contained expression (TMP) and differential gene expression (hi vs lo; E16.5 vs E18.5; L2FC, p-value) for hiloRNA-seq, and expression (TMP) and transcript enrichment (input vs IP; L2FC, p-value) for RIP-seq. The data tables were queried in SQL using expression, differential expression, and significance cutoffs (TMP>5, TMP>10, L2FC>1, L2FC<-1, p-value ≤0.05) to produce lists of noteworthy genes. The resultant gene lists were cross referenced for gene ontology and pathway analysis in g:Profiler [44] and PANTHER [89, 90]. As a secondary approach, full gene lists for specific biological processes were downloaded from the Gene Ontology Consortium [91, 92]. These full gene lists were cross-referenced with our data tables using expression, differential expression, and significance cutoffs filters to identify genes related to a specific biological process that were also significant in one or both data sets. As a final step we also cross-referenced full gene lists from the Gene Ontology Consortium in a three-way analysis that simultaneously queried hilo-RNA-seq, RIP-seq, and our previously published $Dnd1^{Ter}$-seq data sets (GSE132719) [23].

## Data processing, statistical analysis, and Visualization

All software used in the analysis is listed in Table 2. Computing heavy data processing was exclusively performed with SLURM on the Duke Computing Cluster and the Scientific Computing Unit at Weill Cornell Medicine. Computing light data processing was exclusively performed in RStudio on a personal computer [93]. Statistical analysis was performed using the included algorithms that come standard with Seurat (scRNA-seq), and DeSeq2 (RNA-seq) [83, 86]. Initial data visualization utilized included packages. Advanced downstream visualization was either done with EulerR [94, 95] or in GraphPad Prism (version 9.3, La Jolla, CA, USA). Data are presented as the mean ± standard error of the mean. Differences were considered significant when the p-value was ≤ 0.05 (*), ≤ 0.01 (**), ≤ 0.001(***), or ≤ 0.0001 (****). GO identification numbers for GO terms used to derive gene lists are reported in Table 3. Full data tables for hiloRNA-seq and RIP-seq data were also exported together into excel where a rudimentary lookup function accessibly queries the data on a per gene basis and plots basic visualizations of the resultant data (S4 Table). For most final data figures the Ito-Okabe Color Universal Design (CUD) color palette was used for visual inclusivity [96, 97].

**Table 2. Software.**

| Program | Version | Reference |
|---|---|---|
| Biobase | 2.52.0 | [98] |
| BiocGenerics | 0.38.0 | [98] |
| 10x Genomics Cell Ranger | 3.1.0 | [82] |
| clustree | 0.4.4 | [84] |
| corrplot | 0.90 | [99] |
| DBI | 1.1.1 | [100] |
| dbplyr | 2.1.1 | [101] |
| DelayedArray | 0.18.0 | [102] |
| DESeq2 | 1.32.0 | [86] |
| dplyr | 1.0.7 | [103] |
| eulerr | 6.1.1 | [94, 95] |
| ggplot2 | 3.3.5 | [104] |
| gprofiler2 | 0.2.1 | [44] |
| janitor | 2.1.0 | [105] |
| knitr | 2.0 | [106] |
| mixtools | 1.2.0 | [107] |
| openxlsx | 4.2.4 | [108] |
| patchwork | 1.1.1 | [109] |
| plyr | 1.8.6 | [110] |
| R | 4.1.1 | [111] |
| RColorBrewer | 1.1.2 | [112] |
| readr | 2.0.2 | [113] |
| RSQLite | 2.2.8 | [87] |
| Rstudio | 2021.09.1 | [93] |
| Salmon | 1.3.0 | [85] |
| Seurat | 4.0.5 | [83] |
| SRAtoolKit | 2.9.6 | [81] |
| stats | 4.1.1 | [111] |
| stringr | 1.4.0 | [114] |
| tidyverse | 1.3.1 | [115] |
| tximport | 1.20.0 | [116] |
| umap | 0.2.7.0 | [117] |
| viridisLite | 0.4.0 | [118] |
| 10x Genomics Loupe Browser | 4.1.0 | 10x Genomics, proprietary |

## Annexin V Analyses of phospholipid phosphatidylserine

Fetal E16.5 DND1$^{GFP}$ testes were collected as described in the RIP-RNA Extraction method section. However, after TrypLE treatment, testes were dissociated in 200μL of 1x Annexin-binding buffer (Annexin-binding buffer, 5x concentrate, Invitrogen, #V13246). For a GFP-negative control, a E16.5 DND1$^{GFP}$ fetal limb was dissociated using the same method. One drop of Annexin V, Alexa Fluor 647 Ready Flow Reagent (Invitrogen, #R37175) was placed in each sample and incubated in the dark for 15 minutes. Following incubation, the cell suspensions were passed through a 0.32μm cell strainer and were analyzed on a SH800 Cell Sorter by Sony Biotechnology with a 100-μm microfluidic sorting chip and using Fluorescent Laser 2 (FL2) and Fluorescent Laser 4 (FL4) for detection of DND1-GFP and Annexin-647/APC-A, respectively.

**Table 3. GO Biological Process Gene Lists.**

| GO-BP Name | GO:# |
|---|---|
| Histone Modification | GO:0016570 |
| Regulation Chromatin Organization | GO:1902275 |
| Glycolysis | GO:0061621 |
| Oxidative Phosphorylation | GO:0006119 |
| Translation | GO:0006412 |
| RNA binding | GO:0003723 |
| Ribosomal Subunit | GO:0044391 |
| DNA Damage Checkpoint Signaling | GO:0000077 |
| Cell Cycle | GO:0007049 |
| Extracellular Matrix Organization | GO:0030198 |
| Regulation of Histone Ubiquitination | GO:0033182 |
| Golgi Vesicle Transport | GO:0048193 |
| Cytoskeleton Organization | GO:0007010 |
| Chromosome Organization | GO:0051276 |
| DNA Conformation Change | GO:0071103 |
| Protein Modification Process | GO:0036211 |
| Cell Differentiation | GO:0030154 |
| Intrinsic Apoptotic Signaling Pathway | GO:0097193 |
| Extrinsic Apoptotic Signaling Pathway | GO:0097191 |

To determine the geometric mean of Annexin-647/APC-A fluorescence for DND1-GFP-hi and DND1-GFP-lo populations, we first calculated laser compensation using FlowJ v10.8.1 Software (BD Life Sciences). Next, we gated to exclude dead cells ("Live" population, S13A Fig) by using the FSC-A and SSC-A parameters. Taking this population, we identified the DND1-GFP-lo and DND1-GFP-hi populations using the CompFL2A::EGFP-A (x-axis) and cell count parameters (y-axis). GFP fluorescent ranges of $10^4$–$10^{4.6}$ and $10^{4.7}$–$10^{5.1}$ defined the DND1-GFP-lo and -hi populations, respectively. The geometric mean of Annexin V-647 fluorescence was determined for each population.

To determine the percent of cells that are Annexin V-647 positive/negative in each population, we first plotted EGFP fluorescence by the 647/APC-A fluorescence (x and y axes, respectively, S13B Fig), identifying 647/APC-A positive populations (fluorescence of $>10^{3.6}$). After creating this 647/APC-A positive threshold and isolating only the DND1-GFP-hi and DND1-GFP-lo cells, quadrants were created to identify the cells that were DND1-GFP-lo and Annexin V positive (quadrant 1), DND1-GFP-hi and Annexin V positive (quadrant 2), DND1-GFP-hi and Annexin V negative (quadrant 3), and DND1-GFP-lo and Annexin V negative (quadrant 4). Cell counts within these quadrants were then used to for subsequent calculations.

### Reverse-transcription quantitative PCR (RT-qPCR) of *Dnd1*$^{GFP/+}$ DND1-GFP-lo and DND1-GFP-hi populations

Fetal E16.5 *Dnd1*$^{GFP/+}$ testes were collected from a CD1 female mated with a *Dnd1*$^{GFP/GFP}$ male and were dissociated into a single cell suspension as described in the RIP-RNA Extraction method section. The cell suspensions were passed through a 0.32μm cell strainer DND1-GFP-lo and DND1-GFP-hi and cells were collected on a SH800 Cell Sorter by Sony Biotechnology with a 100-μm microfluidic sorting chip using Fluorescent Laser 2 (FL2) for detection of DND1-GFP. Similar to Annexin V analyses, GFP fluorescent ranges of $10^4$–$10^{4.6}$ and $10^{4.7}$–

**Table 4. RT-qPCR Primers.**

| Gene | Direction | Sequence | Reference |
|------|-----------|----------|-----------|
| *eGfp* | Forward | CTT CTT CAA GTC CGC CAT GCC | [27] |
| *eGfp* | Reverse | GGT GTT CTG CTG GTA GTG GTC | [27] |
| *Gapdh* | Forward | GGT GAA GGT CGG TGT GAA CG | [5, 27] |
| *Gapdh* | Reverse | CTC GCT CCT GGA AGA TGG TG | [5, 27] |
| *Dnd1* | Forward | GCC CTG GTA GAA GGT CAG TCA C | [27] |
| *Dnd1* | Reverse | GCC CTG TTC CTA AAC ACT TGG TC | [27] |

$10^{5.1}$ defined the DND1-GFP-lo and -hi populations, respectively. Cells were pelleted by centrifugation at 500 x g for 10 minutes in 4°C, and RNA was extracted using RNeasy Micro Kit per the manufacturer's protocol (Qiagen, catalog #74004). RNA was reverse transcribed into cDNA using the Verso cDNA synthesis kit (Thermo Fisher Scientific, catalog #AB4100A). 1μL of cDNA was used in three separate 10μL PCR reactions, as previously described [27], with primers for DND1, eGFP, and GAPDH, respectively (see below for primer sequences). The PCR program is as follows: initial denaturation for 3 minutes at 95°C followed by 35 cycles of 30 s at 95°C, 30 s at 58°C, and 30 s at 72°C, with a final extension of 5 min at 72°C. RT-PCR products were run on 2.5% agarose gel for 1 hour at 90 V. RT-qPCR products were visualized using a Bio-Rad Imager and Quantity One software (Bio-Rad Laboratories). To determine mean intensities, product bands were subjected to densitometry analysis using ImageJ [75]. For each sample, 4 technical replicates were performed, and biological replicates of ≥4 fetal testes were used. Primers have been previously published [27], and sequences are listed in Table 4.

## Supporting information

**S1 Table. Quantitative data from DND1-GFP Fluorescence Associated Cell Sorting (FACS).**
(XLSX)

**S2 Table. Complete lists for all significantly differentially expressed genes, and GO biological process gene list filtering for hiloRNA-seq and RIP-seq.**
(XLSX)

**S3 Table. Complete list of apoptosis and cell death terms, with gene Ensembl IDs from Biological Process Gene Ontology analysis in g:Profiler of E16.5 DND1-GFP-hi and DND1-GFP-lo cells.**
(XLSX)

**S4 Table. Combined DND1-seq (hiloRNA-seq, RIP-seq) dataset with graphic output.**
(XLSX)

**S5 Table. Complete RIP-seq g:Profiler results.**
(XLSX)

**S1 Fig. Fetal male germ cells visibly express low and high levels of DND1-GFP.** Whole mount confocal imaging of unstained freshly dissected **A:** E16.5 and **B:** E18.5 DND1-GFP testes showing heterogenous endogenous DND1-GFP fluorescence. Insets show cord region (white dotted box, a' and b') and rete testis region (yellow dotted box, a" and b"). All scale bars are 200μm. Scale for A and B in B, scale for a', a", b', and b" in b".
(TIF)

**S2 Fig. Quantification of DND1-GFP in E16.5 MGCs. A:** View of fixed MGCs in a testis cord with DND1-GFP-hi cell marked by an arrow and a DND1-GFP-lo cell marked by a wedge. Scale bar = 50μm. **B, C:** Representative images showing a quantified DND1-GFP-hi cell (**B**) and a quantified DND1-GFP-lo cell (**C**) from an E16.5 testis. b', c': DND1-GFP (endogenous signal). b", c": DNA (Hoechst stain). Scale bar = 10μm. **D:** Quantification of DND1-GFP fluorescence in 50 randomly selected DND1-GFP MGCs from multiple testes cords in 2 separate mice. Dashed line demarcates the relative fluorescence boundary between DND1-GFP-lo cells and DND1-GFP-hi cells (below and above the line, respectively). **E:** Mean intensity of *Dnd1* expression relative to *Gapdh* expression in DND1-GFP-lo (blue) and DND1-GFP-hi (yellow) cells from *Dnd1*$^{GFP/+}$ heterozygotes. **F:** Mean intensity of e*Gfp* expression relative to *Dnd1* expression in DND1-GFP-lo and DND1-GFP-hi cells from *Dnd1*$^{GFP/+}$ heterozygotes. E-F: Technical replicates = 4, biological replicates ≥ 4 fetal testes. P value: ** < 0.01, n.s. = not significant.
(TIF)

**S3 Fig. Pearson correlation matrix of transcriptomes from DND1-GFP-lo and DND1-GFP-hi samples at E16.5 and E18.5.** The matrix shows that DND1-GFP-lo and DND1-GFP-hi cells are most similar to biological replicate samples, but all samples of male germ cells share high similarity.
(TIF)

**S4 Fig. Genes expressed at higher or lower levels in DND1-GFP-hi cells often show a similar correlation in E16.5 sc-RNA-seq data from Law et al.** Re-analysis of E16.5 male germ cell scRNA-seq from [31]. Cluster 0 and Cluster 1 have higher levels of *Dnd1* and Cluster 2 has lower levels of *Dnd1* (also Fig 1E). Violin plots showing levels of other germ cell genes differentially expressed between DND1-GFP-lo and DND1-GFP-lo cells in scRNA-seq of E16.5 male germ cell from [31].
(TIF)

**S5 Fig. Markers previously reported to be associated with the apoptotic or male germ cell determination pathway do not consistently correlate with low or high DND1-GFP.** Imaging from whole mount confocal microscopy on E14.5 DND1-GFP testes (**A-C**) and E18.5 DND1-GFP testes (**D-F**) for: ORF1p (**A & D**), TOMM20 (**B & E**), and CPARP (**C & F**) (red). All sections stained for DND1-GFP (GFP, green), germ cell nuclear antigen (GCNA, blue), and DNA (Hoechst, white). Blue triangles mark DND1-GFP-lo and red triangles mark DND1-GFP-hi germ cells. Scales in C and F are 50μm and apply to all images.
(TIF)

**S6 Fig. Examples of genes from PANTHER protein classes "Transcriptional Regulation", "Defense/Immunity", "Transporters", and "Transmembrane Signal Receptors" that were significantly differentially expressed in DND1-GFP-lo compared to DND1-GFP-hi cells at E16.5 and/or E18.5.** hiloRNA-seq expression P-value (DESeq2) between DND1-GFP-lo and DND1-GFP-hi cells at E16.5 and E18.5: not significant (ns), <0.05 (*), <0.01 (**), <0.001 (***), <0.0001 (****).
(TIF)

**S7 Fig. Chromatin modifiers are disproportionately over-expressed in DND1-GFP-hi cells with genes either shared between DND1-GFP-hi cells at E16.5 and E18.5 or exclusive to DND1-GFP-hi cells E16.5.** Euler plot of significant differentially expressed genes that are part of the GO biological process category "regulation of chromatin modification". There are no genes differentially expressed in both E16.5 and E18.5 DND1-GFP-lo cells. Chart shows top 10

genes for each region of the Euler plot (for full list see S2 Table).
(TIF)

**S8 Fig. Input biological replicates are most similar to each other across all time points, while immunoprecipitation (IP) biological replicates are most similar to each other within a time point.** Pearson correlation matrix of input and IP biological replicates at E14.5, E16.5, E18.5 used in RIP-seq with breakouts showing isolated correlations for input and IP biological replicates at each time point.
(TIF)

**S9 Fig. All paired input and immunoprecipitation (IP) biological replicates at all time points show a subset of transcripts that graph as targets. A:** Scatter plots of log2(TPM) for paired input and immunoprecipitation (IP) biological replicates at E14.5, E16.5, E18.5. Guideline with slope of 1 is plotted to help delineate genes in quadrant I that are above the line, indicative of being a transcript target of DND1. **B:** Gene expression values (TPM) for input and IP biological replicates at E14.5, E16.5, E18.5 for highly expressed housekeeping genes that did not enrich as targets (*Gapdh*, *Acta2*), and a highly expressed gene that did enrich as a target (*Lamb1*).
(TIF)

**S10 Fig. Many epigenetic regulators that are differentially expressed dependent on the level of DND1 are both targets of DND1 and fail to up-regulate in *Dnd1*$^{Ter/Ter}$ mutant germ cells. A:** Heat map of epigenetic regulators showing: L2FC in DND1-GFP-hi vs DND1-GFP-lo cells from hiloRNA-seq (E16.5, E18.5); enrichment of transcript as a DND1 target as a function of L2FC in RIP-seq (E14.5, E16.5, E18.5); and L2FC in *Dnd1*$^{Ter/Ter}$ mutant vs wild type germ cells from *Dnd1*$^{Ter}$-seq (E14.5) from [23]. **B:** Expanded gene-level hiloRNA-seq and RIP-seq data for representatives from each epigenetic regulator family in Figs 4B and S10A. With the exception of *Hdac4*, transcript levels for all genes shown are higher in DND1-GFP-hi cells, and all genes except *Brdt* map as significant targets of DND1 at one or more stages. hiloRNA-seq expression P-value (DESeq2) between DND1-GFP-lo and DND1-GFP-hi cells at E16.5 and E18.5 and RIP-seq enrichment P-value (DESeq2): not significant (ns), <0.05 (*), <0.01 (**), <0.001 (***), <0.0001 (****).
(TIF)

**S11 Fig. Collagens and genes associated with the novel intermediate pro-spermatogonia identity [16, 55] are DND1 targets, but only some are significantly differentially regulated in DND1-GFP-lo and DND1-GFP-hi cells. A:** Heat map of collagens showing L2FC in DND1-GFP-lo vs DND1-GFP-hi cells from hiloRNA-seq (E16.5, E18.5). **B:** Enrichment of transcript as DND1 targets as a function of L2FC in RIP-seq (E14.5, E16.5, E18.5). **C, D**: Expanded gene-level hiloRNA-seq and RIP-seq data for representative genes associated with (**C**) intermediate pro-spermatogonia identity as reported in [55], and (**D**) pro-spermatogonia to spermatogonial stem cell transition as reported in [16]. hiloRNA-seq expression P-value (DESeq2) between DND1-GFP-lo and DND1-GFP-hi cells at E16.5 and E18.5 and RIP-seq enrichment P-value (DESeq2): not significant (ns), <0.05 (*), <0.01 (**), <0.001 (***), <0.0001 (****).
(TIF)

**S12 Fig. Y-linked and X-linked genes are differentially regulated in DND1-GFP-lo and DND1-GFP-hi cells and some enrich as DND1 targets.** Gene-level hiloRNA-seq and RIP-seq data for Y-linked (*Eif2s3y*, *Uty*, *Smcy*) and X-linked (*Eif2s3x*, *Utx*, *Smcx*) genes. hiloRNA-seq expression P-value (DESeq2) between DND1-GFP-lo and DND1-GFP-hi cells at E16.5 and E18.5 and RIP-seq enrichment P-value (DESeq2): not significant (ns), <0.05 (*), <0.01

(**), <0.001 (***), <0.0001 (****).
(TIF)

**S13 Fig. Workflow of DND1-GFP and Annexin V FACS analysis. A:** Live cells were gated using FSC-A and SSC-A parameters, and DND1-GFP-lo cells and DND1-GFP-hi cells were identified with GFP fluorescence and cell count parameters. DND1-GFP-lo and DND1-GFP-hi populations were plotted by 647/APC-A (Annexin) fluorescence. **B:** Annexin V-positive threshold was determined by examining live cells for GFP and 647/APC-A fluorescence. Using this threshold, the DND1-GFP positive cells were divided into DND1-GFP-hi and DND1-GFP-lo, creating four quadrants.
(TIF)

## Acknowledgments

We would like to thank the teams in the: Duke Light Microscopy Core Facility (NIH Shared Instrumentation grant 1S10RR027867-01), especially Dr. Lisa Cameron and Dr. Benjamin Carlson; Duke Cancer Institute Flow Cytometry Shared Resource, especially Dr. Mike Cook and Lynn Martinek; and Duke Center for Genomic and Computational Biology's Sequencing and Genomic Technologies Shared Resource, especially Dr. Nicolas Devos. We are also grateful to past and present members of the Capel Lab, especially Jordan Batchvarov. The computations in this paper were run on HPC resources supported by the Duke Compute Cluster at Duke University and the Scientific Computing Unit at Weill Cornell Medicine.

## Author Contributions

**Conceptualization:** Victor A. Ruthig, Blanche Capel.

**Data curation:** Victor A. Ruthig.

**Formal analysis:** Victor A. Ruthig, Talia Hatkevich, Josiah Hardy, Chloé Mayère.

**Funding acquisition:** Victor A. Ruthig, Josiah Hardy, Blanche Capel.

**Investigation:** Victor A. Ruthig, Talia Hatkevich, Josiah Hardy, Chloé Mayère.

**Methodology:** Victor A. Ruthig, Talia Hatkevich, Matthew B. Friedersdorf, Chloé Mayère.

**Project administration:** Victor A. Ruthig, Blanche Capel.

**Resources:** Talia Hatkevich, Blanche Capel.

**Software:** Victor A. Ruthig, Talia Hatkevich, Chloé Mayère.

**Supervision:** Blanche Capel.

**Validation:** Matthew B. Friedersdorf, Chloé Mayère, Serge Nef, Jack D. Keene.

**Visualization:** Victor A. Ruthig, Talia Hatkevich, Chloé Mayère.

**Writing – original draft:** Victor A. Ruthig, Blanche Capel.

**Writing – review & editing:** Talia Hatkevich, Matthew B. Friedersdorf, Chloé Mayère, Serge Nef, Jack D. Keene.

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
