## [Decision Letter · Decision Letter 0]

19 Sep 2022

Dear Dr Ruthig,

Thank you very much for submitting your Research Article entitled 'The RNA Binding Protein DND1 Is Elevated in a Subpopulation of Pro-Spermatogonia and Targets Chromatin Modifiers and Translational Machinery during Late Gestation' to PLOS Genetics.

The manuscript was fully evaluated at the editorial level and by independent peer reviewers. The reviewers appreciated the attention to an important problem, but raised some substantial concerns about the current manuscript. Based on the reviews, we will not be able to accept this version of the manuscript, but we would be willing to review a much-revised version. We cannot, of course, promise publication at that time.

If you decide to revise the manuscript for further consideration at PLOS Genetics, please aim to resubmit within the next 60 days, unless it will take extra time to address the concerns of the reviewers, in which case we would appreciate an expected resubmission date by email to plosgenetics@plos.org.

[LINK]

Please do not hesitate to contact us if you have any concerns or questions.

Yours sincerely,

Erez Raz

Guest Editor

PLOS Genetics

John Greally

Section Editor

PLOS Genetics

Dear Victor,

I am sending the referees comments on your paper.

Everyone agreed the work is of interest for readers of the Journal and that the data will serve as an important resource.

Two points were considered more important -

1. Provide clear criteria for defining cells as Hi or Lo when microscopy data is concerned (when co-staining with other markers). Was there a threshold above which cells were defined as "Hi"? Was there a control for the signal level at the same imaging plane?

2. A clear suggestion for a "take-home message" should be included. i.e., what is the functional significance of the Hi and Lo Dnd1-expressing populations for gamete formation?

We hope you find the comments of the referees useful !

Reviewer's Responses to Questions

**Comments to the Authors:**

Reviewer #1: Overall

In this very well written manuscript, Ruthig et al identify populations of embryonic germ cells expressing either high or low levels of a Dnd1-GFP reporter knock-in allele. Using this allele, the authors identify fundamental transcriptional differences between these populations and direct RNA targets of the DND1 RNA binding protein. Several regulators of the epigenome and translational machinery are expressed at higher levels in Dnd1-hi cells and are shown to be direct targets of Dnd1. Overall, the experiments shown are very well thought out, performed appropriately, and data correctly interpreted. There are a couple of critical issues to address. First, there is an early emphasis in the abstract and introduction on the role of Dnd1 in teratoma formation in mice based on experiments using the Dnd1-ter allele (with little mention of germ cell deficiency in all background or that teratomas are specific to one background). Although these hi and low cells might tell us something about teratoma formation experiments testing a functional connection are not performed and a teratoma susceptible background is not used. Teratomas are barely mentioned in the discussion. It is suggested that this early emphasis on teratomas be reduced or additional experiments be performed (likely outside the scope of the paper and not being requested). Second, and this addresses overall impact, it is still a bit unclear to me what the role of high Dnd1 during the time course of germ cell development is and where the Dnd1-hi population of cells sits in this process (from PGC to committed pro-spermatogonia). Is this a dynamic population, and representing the nexus at which differentiation is starting/completing? What is the critical nature of this event/stage of a germ cell? Why are pluripotency genes (e.g. Sox2) and critical male germ cell differentiation genes (e.g. Nanos2) expressed higher in these cells compared to Dnd1-low? I think the authors attempt to address this in the discussion but it does need some clarification as a hypothesis. Showing functional impact of high DND1 on germ cells (e.g. DNMT3L or SOX2 protein is detected more frequently or is higher in this population; cells are proliferating/not proliferating) would add significance if achievable.

Major comments

Abstract, introduction, and discussion: There is a lot of early emphasis on loss of Dnd1 and teratomas in mice. Couple of issues with this. First, the experiments conducted are not on a teratoma susceptible background and it isnt mentioned anywhere that teratomas happen on a specific inbred background. Second, there is little said in the discussion tying the data to teratoma development. Third, the one mention of teratomas in the discussion is that de-regulation of epigenetic regulators in the absence of Dnd1 could lead to acquisition of somatic fate and teratomas. How does this work given that the first step in teratoma formation in mice is establishment of pluripotent embryonal carcinoma cells, in which somatic fate has yet to be established? I’d suggest de-emphasizing teratomas in the introduction and focusing on germ cell development/differentiation. Otherwise, data from a teratoma susceptible background – which is outside the scope of the paper (and data I am not asking for) – should be included.

Introduction, line 62-64 – again, teratoma forming capacity of E12.5 germ cells is limited to one strain. So differentiation pathways are “open” in this strain (using this data). Better data to show the general “openness” of these pathways in germ cells in general is the capacity to generate embryonic germ cell lines from early germ cells and that capacity being lost as sex-specific differentiation initiates.

Introduction - Maybe discuss the (putative) molecular functions of Dnd1 up front. Doesn’t have to be a lot of detail (can leave for the discussion to integrate the findings of the paper into its known role), but something should be mentioned in the intro.

Introduction – what experimental evidence is there that mouse germ cells “trans-differentiate” to form teratomas? I think that this is a point of contention in the field – are germ cells lineage specified prior to the time tumors form. Do they “escape” germ cell identity or fail to acquire/lock in identity. I’d suggest not using this term, provide specific evidence that they “trans-differentiate”, or indicate that the precise mechanism of germ cell transformation/trans-differentiation/de-dedifferentiation is not known.

Abstract, introduction, and discussion: The potential critical role of translational control/rates in teratoma formation from germ cells has been previously described (Heaney et al, Hum Mol Genetics, 2009). This work describes deficiency for Eif2s2 as reducing teratoma incidence in mice. Important considering that the authors address DND1 regulation of translation regulators/factors and specifically mention Eif2s3x and y). So, if teratomas are going to be part of the theme of the manuscript, it would be a good idea to reference this paper. These data are not in agreement with the concept of the role of Dnd1 in regulation of translation factors and teratomas. Dnd1-hi cells have higher expression of translation regulators (assuming higher rates of translation?) but should also maybe be more protected from forming teratomas? Dnd1 deficient cells should then maybe have less expression of these translation regulators but Dnd1-ter germ cells form embryonal carcinoma cells at a high rate in 129 mice.

Results: Data in Fig S1 is very interesting. Is the GFP weaker in areas closest to the rete testis? Is there some biological significance to this? Or is this an imaging artifact?

Results: Fig 1 - Does expression from the endogenous (unmodified) Dnd1 allele match the reporter? I’d assume that one could collect the hi and low cells and look at RNA expression in hets? Probably a better control to assess “transgene” artifact than going to independent RNA-seq studies. You have the hi and low populations separated here. Should be able to confirm native allele RNA expression. But, looks like dnd1 expression in the sorted populations might be embedded in the RNA-seq data (Fig 3C)? Might want to find a way of connecting these.

Results: Fig 2 – Using qualitative fluorescent intensity of the GFP reporter to look at whether hi and low GFP cells associate with these markers or not is probably not a great approach. I see the value of these data, but the RNa-seq data are much more convincing. I suggest moving to supporting info. There isnt much gained from these data.

Results: Fig 3D – not too sure what is being shown here. Is this a combination of different genes? If yes, this needs to be explained a bit better in the results (and which genes are involved).

Results: Line 200 – 202 “Despite the convergence of some transcripts, many epigenetic regulators maintain higher expression levels (TPM≥5.0, log2-fold change (L2FC)≥1.0, p-val<0.05) in DND1-GFP-hi cells during G0.” Can it really be said that these cells are in G0. I don’t think the proliferative status (or stage of cell cycle arrest – pH3 vs Ki67 staining) of the hi and low cells were assessed here. Is this known?

Results and discussion: After reading the discussion, I was still left with the questions – what is the fundamental difference between Ddn1-hi and low? Other than differences in gene expression – how are they functionally different. Proliferating/not proliferating? Still expressing pluripotency factors (protein)/not retain pluripotent characteristics? Initiated NANOS2 or DNMT3 (a or l) protein expression or not? For example, Dnmt3a and Dnmt3l RNA are expressed higher in DND1-hi cells. How does this translate to protein? Are the proteins differentially present/absent in the DND1-low and hi populations? Maybe can do this by imaging? Are the proteins expressed differently? Understand that this is much harder as westerns would be terribly difficult with cell numbers available.

Results: Most of the direct targets of DND1 appear to be expressed higher in DND1-hi cells. Are there examples of transcripts that are direct targets of DND1 and are lower in DND1-hi cells? Gets to an overall.

Discussion: Line 341 – 344. Nanos2 is not a marker of pluripotency. In fact, having some genes such as Sox2 (a marker of pluripotency) and a male germ regulator of sex-specific differentiation (Nanos2) both increased in expression in Dnd1-hi cells is counterintuitive. This should be addressed in the discussion. Other genes that are indicative of male germ cell differentiation (Tdrd’s, methyltransferases) are also higher. So why would Sox2, Tfap2c be higher? Are these cells that are being actively transitioned from a more “PGC-like” state to a more “differentiated” state so we are seeing this in the transcriptional profiles? Maybe I missed it, but is Sox2 a target of Dnd1? If not, would support that these Dnd1-hi cells are being actively pushed to differentiate by high Dnd1?

Minor comments

Overall, the figures are put together very well. Recommend that the text sizing be adjusted for some panels (e.g. 3C, 7B) as they are very hard to read without zooming in A LOT.

Results: Probably best to refer to the allele as a knock-in, not a transgene which usually is used when describing a randomly integrated DNA construct.

Reviewer #2: The manuscript by Ruthig et al., describes molecular phenotyping analysis of a subpopulation of male germ cells in mouse, which has been identified by transgenic insertion of GFP into DND1 gene and detection of a selectively high DND1 expressing cell population. The discovery of this cell population by the transgene is validated by endogenous DND1 expression analysis in single cell transcriptome and further analysed by differential gene expression analysis. This analysis suggest striking heterogeneity and associated differences in gene expression profiles between the two cell populations. Differentially expressed genes suggest DND1-associated regulation of cell cycle, cellular metabolism, protein translation and chromatin state regulator genes. Of note, several of these functions were already suggested before in the literature. Subsequently, the authors identify RNAs bound by DND1 using RIP and suggest that the two datasets indicate roles for DND1 in regulating subsets of RNAs and affect translation of target genes. The authors main claim is that the different DND1 levels indicate male germ cell heterogeneity, and their results pinpoint features of the heterogenous cell populatjons using their transcriptomic and RIP analyses. The manuscript is primarily descriptive but may represent potentially important, biologically relevant description of germ cell heterogeneity and may contribute to better understanding of the role of DND1 in germ cell fate if the heterogeneity detected is better linked to more comprehensive cell biological/histological characterisation of the novel cell population and if the transcriptome and RIP results are more robustly correlated.

There are several questions arising from the current version of the ms, which need addressing (particularly Q1-3).

1. Does the heterogeneity detected by the transgenic insertion of GFP marker faithfully represent true endogenous (non-transgenic) heterogeneity? While single cell data suggest that high expressing DND1 cells indeed exist, are those cells in the single cell data showing similar transcript dynamic of candidate targets and DEGs seen in the FACS isolated cells?

2. What is the cellular and functional relevance of the molecular heterogeneity detected in this ms? This question is important for assessing the meaning of the detected transcriptomic differences, given that the authors do not see correlation between the high DND1 expression and known variable features of germ cells such as differentiating or apoptotic germ cells. Without providing some validation to the observed DEG and RNA targeting phenomena the study will not represent sufficient advance. A complementing histological characterisation of the high DND1 cells will also offer the benefits of validation of the predicted molecular (e.g. chromatin remodelling) and cellular (e.g. cell cycle) functions highlighted by the authors. Such characterisation/validation may include high DND1 cell group-specific detection of changes in cell cycle markers, detection of expected global changes in repressive chromatin states (e.g. comparative immunohistochemistry of DNA and histone methylation markers in high and low DND1 cells), analysis of global translation regulation and oxidative phosphorylation.

3. A related question to that above, is whether the authors see differential subcellular localisation patterns of DND1 and or cell distributions at different stages of germ cell development between DND1 high and low cells? Is there any change in cellular behaviour that may indicate the distinct fates between these cell groups? Figure 3 indicates some clustering of DND1 high cells. Is this stereotypical for tissue position or frequency of clustering?

4. It is difficult to reconcile or appreciate the partial overlap between DEG and RIP results. The composition of the subsets of genes that are differentially expressed in high DND1 cells that appear to be direct targets could be explained by the composite effect of DND1 dependent stabilisation/degradation of RNAs and the indirect effect of DND1-dependent translation regulation associated RNA degradation. One would expect tighter correlation between temporal expression dynamic (DEG) and temporal degree of DND1 targeting (RIP) at the different stages of germ cell development tested upon direct effect. Can such temporal analysis of correlation between the expression changes and RIP results in E16.5-E18.5 be carried out?

5. Not essential but certainly relevant question and relatively easy to check whether the DND1 targets match those which are dependent on DND1 homologs in other organisms (e.g. zebrafish).

Minor points:

1. The data in Figure 4 is hard to follow due to the multiple colours which are difficult to assign using the colour code in the legends. Can the authors revise this figure to link the DEG categories shown in stack bars across stages and show the GO terms next to the stacks?

2. Font size in Fig. 7B should be increased.

3. Protein is misspelt in Figure 4 legend.

4. There is no need to abbreviate the word Figure in Figure titles (or indeed throughout the ms).

Reviewer #3: Ruthig et al report on two populations of male germ cells characterized by different levels of Dead end expression. The difference between the two population is detected by the expression level of a fusion between DND1 and GFP that is inserted within the endogenous locus. The cell population expressing higher level of DND1-GFP show higher level of epigenetic regulators RNAs, and the protein binds RNAs of such regulators.

As DND plays central roles in the development of germ cells, the lists of possible DND1 in vivo targets are important resource for future functional studies. The results are thus clearly of interest for the readership of the Journal.

Addressing/explaining better some of the issues listed below would improve the manuscript.

Introduction

1. “During the last third of fetal life until a few days after birth (E14.5 to about P2) (7-9), male germ cells (MGCs) enter a long period of cell cycle arrest, during which they undergo extensive reprogramming that reduces their potential to give rise to teratomas and drives cells towards the pro-spermatogonial (PSG) fate.” – has it been critically shown that “extensive reprogramming” at these stages indeed reduces teratoma formation? If so, the source should be cited. In certain contexts, reprogramming is thought to promote teratoma formation, so would be good if the authors define “reprogramming” in their case and align them with previous literature (e.g. https://www.nature.com/articles/nature12586 ).

2. “Dead end 1 (Dnd1), an RNA binding protein (RBP), underlies the classic mouse mutation “Ter”, named for the high incidence of teratomas in male Dnd1Ter/Ter mutants” – mention that this holds true for a specific genetic background only.

3. “loss or mutation of Dnd1 leads to a severe loss of embryonic germ cells soon after their specification (17, 18), and in zebrafish, transformation of germ cells to somatic lineages” – not clear what “severe loss” means. Would “loss“ be sufficient. In addition, the way it is phrased, it sounds as if the apparent loss of cells in the mouse is not (also) a result of cell transformation. Is cell death the only way of losing the germ cells in Ter mice?

4. “Although some of these chromatin modifiers mapped as mRNA targets of DND1 in HEK293 cells, many were not expressed in this cell line” – This sentence is clear for those who read reference 21. I would add 2 sentences, explaining the results presented in references 20 and 21 such that this part of the introduction is a “stand alone” paragraph.

Results

5. “At E14.5, the DND1-GFP-hi population accounts for <1% of MGCs. Their numbers peak at E16.5, when DND1-GFP-hi cells represent 12% of total MGCs, and decline again at E18.5, when the DND1-GFP-hi population accounts for only 5% of MGCs. “. In table S2, the numbers and percentages do not add up properly (are not always 100%, the total MGCs is less/more than hi + lo). Is there an explanation for that?

6. What stage is presented in Fig 1A?

7. In S1 E16, what should one consider as hi and lo ? there are regions that appear to be devoid of signal, some lo and some hi. The definition, of hi and lo should be made clear here. As presented, it seems like there are domains of hi on either side of the cords and that they are larger than what one would have expect from a 12% population size.

8. Define GCNA in Fig 2. What does the White “DNA” in 2A-C stand for? Why not show the level of these specific markers derived from the RNA seq done at E16?

9. The authors should try to exclude the option that the hi and lo are not a result of small differences in the rate of development among cells at E16.5-E18.5, with all the cells becoming similar to each other at P3 (perhaps hi). It is not obvious how can one compare the lo in P3 with that in E18.5. If it is based on absolute GFP level it could signify more uniform distribution of cell states at the E14.5 and P3 -> It would help if the authors present arguments that make this option (that the small hi population is not simply composed of “faster”, or “slower” cells) less likely.

10. Conducting analysis similar to that presented in Fig 2 for 1-2 differentially-expressed markers derived from Fig 3 (antibody as in 2, or employing a quantitative in situ hybridization method) would increase the confidence in the results presented in both figures – i.e., show the correlations shown in 3 following the experimental scheme of Figure 2.

11. What justifies presenting the hi and lo of different stages within the same graphs in Figure 6? Were the counts normalized to the expression level of a house-keeping gene for example?

12. The ideal control for the RIP would have been a GFP-tagged mutated Dnd1, but such a knockin is obviously not available. A control of RIP using another GFP fusion could have helped here. Perhaps I missed it, but what would increase the confidence in the RIP results and show specificity would be a demonstration of lack of very good (not just a few examples provided) correlation between the normalized expression level of transcripts (not rRNAs which are not polyadenylated) and the RIP results. The authors should just present the results more clearly to exclude this option.

13.

“ Nearly all of these chromatin regulators are genes that fail to be up-regulated in DND1Ter/Ter mutant germ cells by E14.5 “. Here the authors focus on RNAs encoding for proteins that are more “interesting” in the context of this work. How do other genes behave in this comparison?

Discussion

14. Is there anything special in germ cells at the stages studied that could account for the high representation of vesicle transport encoding RNAs?

**Have all data underlying the figures and results presented in the manuscript been provided?**

Reviewer #1: Yes

Reviewer #2: Yes

Reviewer #3: Yes

PLOS authors have the option to publish the peer review history of their article (what does this mean?). If published, this will include your full peer review and any attached files.

Reviewer #1: No

Reviewer #2: No

Reviewer #3: No

---

## [Decision Letter · Decision Letter 1]

6 Feb 2023

Dear Dr Ruthig,

We are pleased to inform you that your manuscript entitled "The RNA Binding Protein DND1 Is Elevated in a Subpopulation of Pro-Spermatogonia and Targets Chromatin Modifiers and Translational Machinery During Late Gestation" has been editorially accepted for publication in PLOS Genetics. Congratulations!

Yours sincerely,

Erez Raz

Guest Editor

PLOS Genetics

John Greally

Section Editor

PLOS Genetics

Comments from the reviewers (if applicable):

Dear Victor, Dear Blanche,

As you can see in the comments, everyone agreed that this version of the manuscript should be published in the PLOS Genetics.

One of the referees had a few very minor additional points, which I am sure should not be a problem to address.

Congratulations and apologies for the long time it took.

Erez

Reviewer's Responses to Questions

**Comments to the Authors:**

Reviewer #1: In this revised manuscript, Ruthig et al identify populations of embryonic germ cells expressing either high or low levels of a Dnd1-GFP reporter knock-in allele. Using this allele, the authors identify fundamental transcriptional differences between these populations and direct RNA targets of the DND1 RNA binding protein. Several regulators of the epigenome and translational machinery are expressed at higher levels in Dnd1-hi cells and are shown to be direct targets of Dnd1.

I commend the authors for their very careful consideration of reviewer comments and the nice revisions they have made to this manuscript. In particular the authors have addressed 2 critical concerns raised out of the reviews. First, the authors used DAPI staining to "normalize" the GFP expression levels in their microscopy data, which has addressed concerns about identifying GFP-hi and GFP-low cells in their images. Second, the authors have added an essential experiment that shows Annexin V staining, a marker of apoptosis, is convincingly more associated with GFL-low cells. This along with the established literature that there is loss of a large proportion of male germ cells around birth, provides interesting biological significance to the 2 populations and the genes that they express. I was also pleased to see additional work performed to verify that endogenous Dnd1 expression correlates with GFP expression levels.

I do not have any major concerns left to be addressed.

A few minor comments:

1. I would add a reference to this statement on line 504 "A large proportion of MGCs are lost by the time of birth." Probably well accepted within the mammalian germ cell community, but outside of that, it is not necessarily common knowledge.

2. Line 103 – still a use of "transgenic" to describe the allele. Knock-in would be better.

3. Typo on Line 248 - "express much higher levels of genes associated with associated with the germline...."

Reviewer #2: This substantially revised ms has addressed all of my queries raised. The replies are satisfactory and the new results, particularly Figure 3 demonstrating striking differences in Annexin V expression suggest phenotypic consequences to distinct DND1 levels in the identified germ cell populations and improved the ms. The new data in Suppl. Fig. 4 indicates correlation between the single cell and bulk transcriptomes making the observations more robust.

Reviewer #3: The authors addressed all the issues I raised, and provided good reasons for not doing so in some cases.

I support publishing this paper without any additional changes.

**Have all data underlying the figures and results presented in the manuscript been provided?**

Reviewer #1: Yes

Reviewer #2: Yes

Reviewer #3: Yes

PLOS authors have the option to publish the peer review history of their article (what does this mean?). If published, this will include your full peer review and any attached files.

Reviewer #1: No

Reviewer #2: No

Reviewer #3: No

**Data Deposition**

http://datadryad.org/submit?journalID=pgenetics&manu=PGENETICS-D-22-00785R1

**Press Queries**

---

## [Editor Report · Acceptance letter]

23 Feb 2023

PGENETICS-D-22-00785R1 

The RNA Binding Protein DND1 is Elevated in a Subpopulation of Pro-Spermatogonia and Targets Chromatin Modifiers and Translational Machinery During Late Gestation 

Dear Dr Ruthig, 

We are pleased to inform you that your manuscript entitled "The RNA Binding Protein DND1 is Elevated in a Subpopulation of Pro-Spermatogonia and Targets Chromatin Modifiers and Translational Machinery During Late Gestation" has been formally accepted for publication in PLOS Genetics! Your manuscript is now with our production department and you will be notified of the publication date in due course.

With kind regards,

Anita Estes

PLOS Genetics

On behalf of:
